# RNA-binding proteins direct myogenic cell fate decisions

Joshua R Wheeler[1,2,3,4,5†], Oscar N Whitney[6†], Thomas O Vogler[2,7,8], Eric D Nguyen[2,9], Bradley Pawlikowski[7], Evan Lester[1,2], Alicia Cutler[7], Tiffany Elston[7], Nicole Dalla Betta[7], Kevin R Parker[10], Kathryn E Yost[10], Hannes Vogel[4], Thomas A Rando[11,12,13], Howard Y Chang[10,14], Aaron M Johnson[9,15], Roy Parker[3]*, Bradley B Olwin[7]*

[1]Department of Biochemistry, University of Colorado, Boulder, United States; [2]Medical Scientist Training Program, University of Colorado Anschutz Medical Campus, Aurora, United States; [3]Howard Hughes Medical Institute, University of Colorado, Boulder, United States; [4]Department of Pathology, Stanford University, Stanford, United States; [5]Department of Neuropathology, Stanford University, Stanford, United States; [6]Department of Molecular and Cell Biology, University of California, Berkeley, Berkeley, United States; [7]Department of Molecular, Cellular and Developmental Biology, University of Colorado, Boulder, United States; [8]Department of Surgery, University of Colorado, Aurora, United States; [9]Molecular Biology Program and Department of Biochemistry and Molecular Genetics, University of Colorado Anschutz Medical Campus, Aurora, United States; [10]Center for Personal and Dynamic Regulomes, Stanford University, Palo Alto, United States; [11]Department of Neurology and Neurological Sciences, Stanford University School of Medicine, Stanford, United States; [12]Paul F. Glenn Center for the Biology of Aging, Stanford University School of Medicine, Stanford, United States; [13]Center for Tissue Regeneration, Repair, and Restoration, Veterans Affairs Palo Alto Health Care System, Palo Alto, United States; [14]Howard Hughes Medical Institute, Stanford University, Stanford, United States; [15]University of Colorado School of Medicine, RNA Bioscience Initiative, University of Colorado Anschutz Medical Campus, Aurora, United States

*For correspondence:
roy.parker@colorado.edu (RP);
olwin@colorado.edu (BBO)

†These authors contributed equally to this work

**Abstract** RNA-binding proteins (RBPs), essential for skeletal muscle regeneration, cause muscle degeneration and neuromuscular disease when mutated. Why mutations in these ubiquitously expressed RBPs orchestrate complex tissue regeneration and direct cell fate decisions in skeletal muscle remains poorly understood. Single-cell RNA-sequencing of regenerating *Mus musculus* skeletal muscle reveals that RBP expression, including the expression of many neuromuscular disease-associated RBPs, is temporally regulated in skeletal muscle stem cells and correlates with specific stages of myogenic differentiation. By combining machine learning with RBP engagement scoring, we discovered that the neuromuscular disease-associated RBP Hnrnpa2b1 is a differentiation-specifying regulator of myogenesis that controls myogenic cell fate transitions during terminal differentiation in mice. The timing of RBP expression specifies cell fate transitions by providing post-transcriptional regulation of messenger RNAs that coordinate stem cell fate decisions during tissue regeneration.

## Editor's evaluation

Wheeler and colleagues examine genetic pathways of myogenesis in regenerating muscle. Using extensive single cell and whole-genome analyses, they uncover a new role for the RNA binding protein hnRNP A2B1, showing that it plays a key role in defining muscle-specific splicing patterns during development.

## Introduction

Skeletal muscle is among the longest-lived tissues in the human body, is essential for locomotion, respiration, and longevity, and thus requires constant maintenance (*Sharples et al., 2015*). Skeletal muscle is comprised of postmitotic myofibers that house resident muscle stem cells (MuSCs), which can repair skeletal muscle following damage (*Lepper et al., 2011*; *Murphy et al., 2011*; *Sambasivan et al., 2011*; *Shi and Garry, 2006*). MuSCs are typically quiescent, but (*Feige et al., 2018*) in response to muscle injury, MuSCs activate, proliferate, and then either self-renew or differentiate and fuse to repair myofibers (*Baghdadi and Tajbakhsh, 2018*; *Sincennes et al., 2016*).

Single-cell analyses of regenerating muscle demonstrate the rich cellular complexity governing myogenesis and make two key observations (*De Micheli et al., 2020*; *Dell'Orso et al., 2019*; *Giordani et al., 2019*). First, in response to damage, MuSCs exit quiescence and progress through a hierarchical, dynamic myogenic program and either commit to terminal differentiation or self-renew and reacquire a quiescent state. Second, as activated MuSCs progress through myogenesis, MuSCs experience dramatic global changes in gene expression (*Barruet et al., 2020*). This rapid change in gene expression requires MuSCs to regulate a vast amount of newly transcribed RNA encoding fate-specifying transcription factors and skeletal muscle contractile apparatus constituents, among other myogenic proteins.

RNA is regulated by an arsenal of abundant RNA-binding proteins (RBPs). Although much of the effort to understand the roles of RBPs in myogenesis has focused on specific RBPs associated with disease (*Apponi et al., 2011*), recent work has identified functions for RBPs not involved in neuromuscular diseases that include (1) maintenance of MuSCs quiescence (*de Morrée et al., 2017*), (2) MuSC activation and expansion (*Cho and Doles, 2017*; *Farina et al., 2012*), (3) myogenic differentiation (*Hausburg et al., 2015*; *Vogler et al., 2018*), and (4) MuSC-self-renewal (*Chenette et al., 2016*; *Hausburg et al., 2015*).

RBPs regulate RNA splicing, where RNA is targeted and bound by RBPs co-transcriptionally to ensure correct splicing of nascent RNA transcripts (*Dassi, 2017*; *Hentze et al., 2018*) where alternative RBP-mediated splicing yields a rich diversity of RNA isotypes critical for translating proteins that regulate myogenesis (*Brinegar et al., 2017*; *Imbriano and Molinari, 2018*; *Nakka et al., 2018*; *Weskamp et al., 2020*). The loss of specific RBPs and the resultant effects on splicing affect MuSC quiescence, MuSC activation, and differentiation; mutations in RBPs are associated with muscular dystrophies and age-related neuromuscular diseases (*Calado et al., 2000*; *Hinkle et al., 2019*; *Xue et al., 2020*; *Yu et al., 2009*). Amyotrophic lateral sclerosis, inclusion body myopathy, and muscular dystrophies are caused by RBP mutations, leading to progressive muscle degeneration (*Lukong et al., 2008*; *Xue et al., 2020*). In these disorders, disease-causing mutations frequently impair RBP splicing function (*Cortese et al., 2014*; *Singh et al., 2018*; *Taylor et al., 2016*). Restoring RBP splicing function prevents or delays the onset of these disorders, identifying splicing dysregulation as a key driver of pathology (*Naryshkin et al., 2014*; *Scotti and Swanson, 2016*; *Wirth et al., 2020*).

Even though RBPs are essential for skeletal muscle regeneration and are frequently mutated in neuromuscular disease, little is known regarding the mechanisms regulating the timing of RBP function during myogenesis or the mechanisms by which RBPs influence myogenic cell fate decisions. Using single-cell RNA-sequencing, we examined temporal RBP expression of several neuromuscular disease-associated RBPs in MuSCs during myogenic differentiation to clarify the timing of RPB expression during muscle regeneration and identify networks of RBPs involved in myogenic cell fate transitions.

## Results

### Myogenic cell fate transitions revealed by single-cell RNA-sequencing

Skeletal muscle has a remarkable ability to repair following injury mediated by MuSCs, which activate, proliferate, and differentiate to produce the majority of myonuclei by 4 days post injury (dpi) (*De Micheli et al., 2020*; *Cuter et al., 2019*). During this time, MuSCs self-renew to replenish the quiescent MuSC stem cell population as demonstrated by DNA-lineage-tracing experiments (*De Micheli et al., 2020*; *Cuter et al., 2019*). MuSCs undergo dramatic transcriptional changes during myogenesis, of which the resultant RNA is regulated by RBPs. Yet, the timing and function of RBPs during myogenesis remain understudied. Here, we sought to define RBP function in individual cells during MuSC activation, proliferation, differentiation, and self-renewal.

We performed single-cell RNA-sequencing on regenerating skeletal muscle at 4 dpi and 7 dpi following an injury by $BaCl_2$ injection into adult mouse tibialis anterior (TA) muscles (*Caldwell et al., 1990*). We observed infiltration of immune cells, dynamic cycling of individual fibro/adipogenic progenitors (FAPs), and a robust myogenic progenitor population in the sequencing datasets (*Figure 1A*, *Figure 1—figure supplement 1A and B*). Sequencing dataset analyses show strong reproducibility amongst biological replicates (*Figure 1—figure supplement 1A and B*) defining the cellular composition of regenerating skeletal muscle. The cellular constituents we identified overlap with recently published muscle single-cell RNA-sequencing datasets (*Barruet et al., 2020*; *De Micheli et al., 2020*; *Dell'Orso et al., 2019*). Thus, acutely injured muscle undergoes similar cellular regeneration irrespective of the inciting injury.

Gene expression and cell cycle scoring analysis of the myogenic cells reveals three dominant myogenic clusters: (i) a proliferative *Pax7*-positive MuSC population, (ii) a differentiating *Myogenin* (Myog)-positive population exiting cell cycle, and (iii) a postmitotic, terminally differentiating muscle population expressing sarcomeric mRNAs (*Figure 1B and C*, *Figure 1—figure supplement 1C*). These dominant myogenic populations are further subclassified into nine subclusters that define the regenerating myogenic population (*Figure 1D*) and may represent cells at various points along a more continuous differentiation trajectory or may reflect specific cell state transition points during myogenesis.

We next identified temporal dynamics between each of the nine myogenic subclusters using RNA velocity (*Bergen et al., 2020*), which infers cellular dynamics for a single cell by comparing the ratio of unspliced, pre-mRNA to mature mRNA (*La Manno et al., 2018*). RNA velocity shows myogenic differentiation directionality within specific subclusters (*Figure 1—figure supplement 1C*), resolving myogenic differentiation trajectories using processed and unprocessed RNA (*Figure 1—figure supplement 1C*). To infer connectivity and directionality between the nine subclusters, we employed partition-based graph abstraction (PAGA), which estimates connectivity among different subclusters while preserving global data topology, providing a more granular analysis than traditional pseudotemporal analyses (*Wolf et al., 2019*). PAGA identified a directed connectivity between the myogenic subclusters 0–6 (*Figure 1E*, *Figure 1—figure supplement 1D*) where we see robust connections between MuSC clusters and differentiating clusters. These connections highlight the inherent complexity of MuSC cell fate decisions (*Figure 1—figure supplement 1D*). Very few connections lead to mature muscle, suggesting that progenitor MuSC fate decisions are plastic. By contrast, the commitment to terminal differentiation may follow a single universal pathway. Together, these data reveal a trajectory map of in vivo myogenesis during muscle regeneration.

### RNA-binding protein expression is temporally defined during myogenesis

The mechanisms of myogenic cell fate change are likely multifactorial and include post-transcriptional regulation of mRNAs. Post-transcriptional regulation permits rapid and dynamic cell fate changes in myogenic cells that is critical for cell fate transitions (*Hausburg et al., 2015*), but is poorly understood during myogenesis at a cellular resolution (*Apponi et al., 2011*; *Hinkle et al., 2019*; *Weskamp et al., 2020*). Since a number of RBP transcripts increase during muscle regeneration, examining the timing and magnitude of expression for RBPs in individual MuSCs may reveal an unexplored role for RBPs in mediating myogenic cell fate transitions (*Weskamp et al., 2020*). We hypothesize that RBP expression timing influences myogenic cell fates by post-transcriptionally regulating RNA. We defined RBP expression in MuSCs, focusing on myogenic subclusters 0–6. The MuSCs in these subclusters

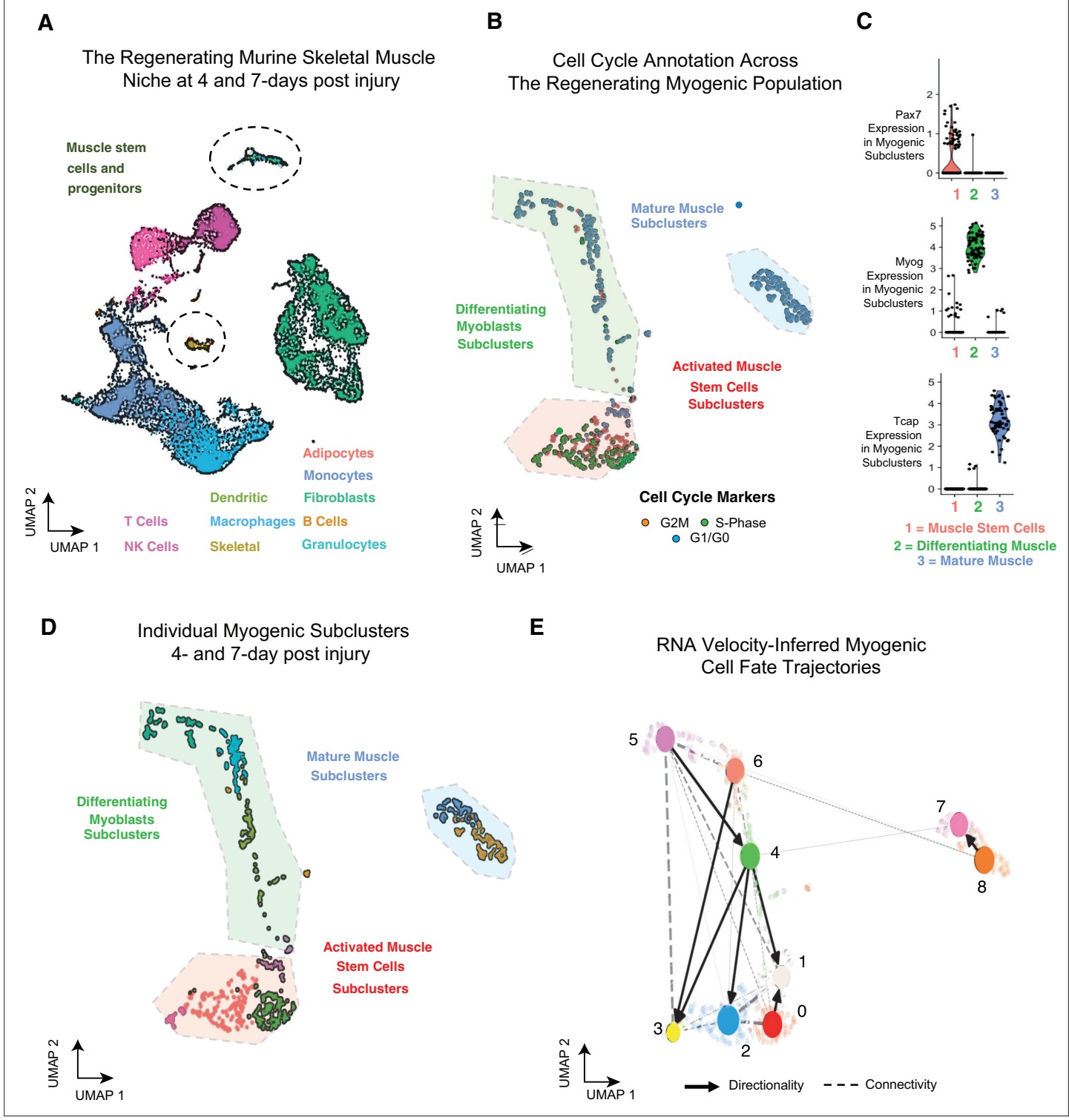

**Figure 1.** Single-cell analysis reveals myogenic cell fate transitions in regenerating skeletal muscle. (**A**) Single-cell atlas of regenerating skeletal muscle at 4 and 7 days post injury (dpi). (**B**) Cell cycle scoring in regenerating myogenic subclusters. (**C**) Violin plots showing expression of myogenic markers of regeneration per myogenic clusters. (**D**) Myogenic subclusters comprising the regenerating myogenic cellular population. (**E**) RNA velocity-inferred myogenic cell fate trajectories. See also *Figure 1—figure supplement 1*.

The online version of this article includes the following figure supplement(s) for figure 1:

**Figure supplement 1.** Related to *Figure 1*.

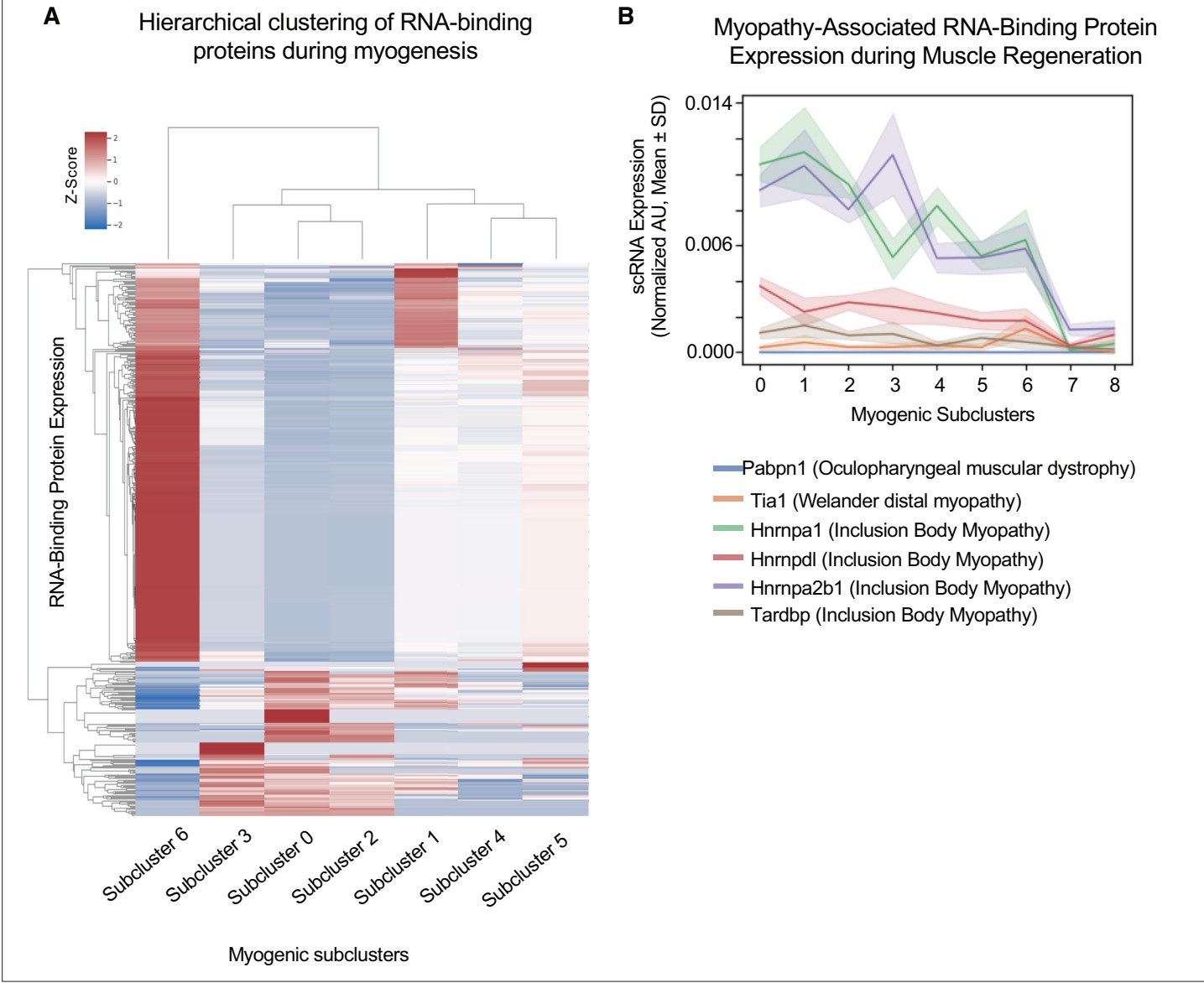

**Figure 2.** Cluster-specific RNA-binding protein expression is temporally defined during myogenesis. (**A**) Hierarchical clustering of RNA-binding protein expression in myogenic subclusters 1–6 during myogenesis. (**B**) Line plot of select myopathy-associated RNA-binding protein expression in subclusters 0–8 during myogenesis.

undergoing cell fate changes to either self-renew, continue to proliferate, or terminally differentiate. Thus, examining RNA regulation in these cells may reveal roles for RBPs in directing cell fate decisions.

We first examined which RBPs are expressed in MuSCs in subclusters 0–6 (*Castello et al., 2012*; *Perez-Perri et al., 2018*) and subsequently performed hierarchical clustering for the expressed RBP transcripts in each subcluster (*Figure 2A*). These results identify networks of RBPs that are either upregulated or downregulated in specific subclusters. Thus, as the transcriptional state of MuSCs changes so too does the expression of RBP networks.

We hypothesized that RBP expression timing relates to the times in which those RBPs are functioning. Thus, dysfunction of RBPs at specific cell states may disrupt cell fate decisions and explain broadly why mutations or dysfunction in many RBPs cause neuromuscular diseases. For example, *Tardbp* (TDP-43) dysfunction impairs myogenic proliferation, whereas loss of *Hnrnpa1* disrupts terminal myogenic differentiation (*Liu et al., 2017*; *Vogler et al., 2018*). Focusing on these disease-associated RBPS, we found that *Tardbp* expression is highest in subcluster 1, and in contrast, *Hnrnpa2b1* expression is highest in subcluster 3, while *Tia1* expression is highest in subcluster 6 (*Figure 2B*). Thus,

distinct disease-associated RBP expression profiles peak in different subclusters, corresponding to different time points during myogenesis and distinct cell fate decisions.

Two of the most highly expressed RBPs are *Hnrnpa1* and *Hnrnpa2b1* (*Figure 2B*), and when disrupted cause inclusion body myopathy (*Kim et al., 2013*). *Hnrnpa1* and *Hnrnpa2b1* expression is highest in select proliferating and differentiating subclusters. While *Hnrnpa1* function is essential for muscle differentiation, *Hnrnpa2b1* function in muscle is unknown (*Li et al., 2020*). Thus, we focused on *Hnrnpa2b1* as a model RBP for understanding cell state-specific RBP function. Defining the role for *Hnrnpa2b1* in specific cell states may provide insight into how disruption of this RBP causes inclusion body myopathy.

### *Hnrnpa2b1* expression dynamics in regenerating skeletal muscle

*Hnrnpa2b1* regulates RNA splicing, stability, and transport (*Alarcón et al., 2015*; *Geissler et al., 2016*; *Percipalle et al., 2002*) and, in stem cells, *Hnrnpa2b1* regulates differentiation (*Wang et al.,*

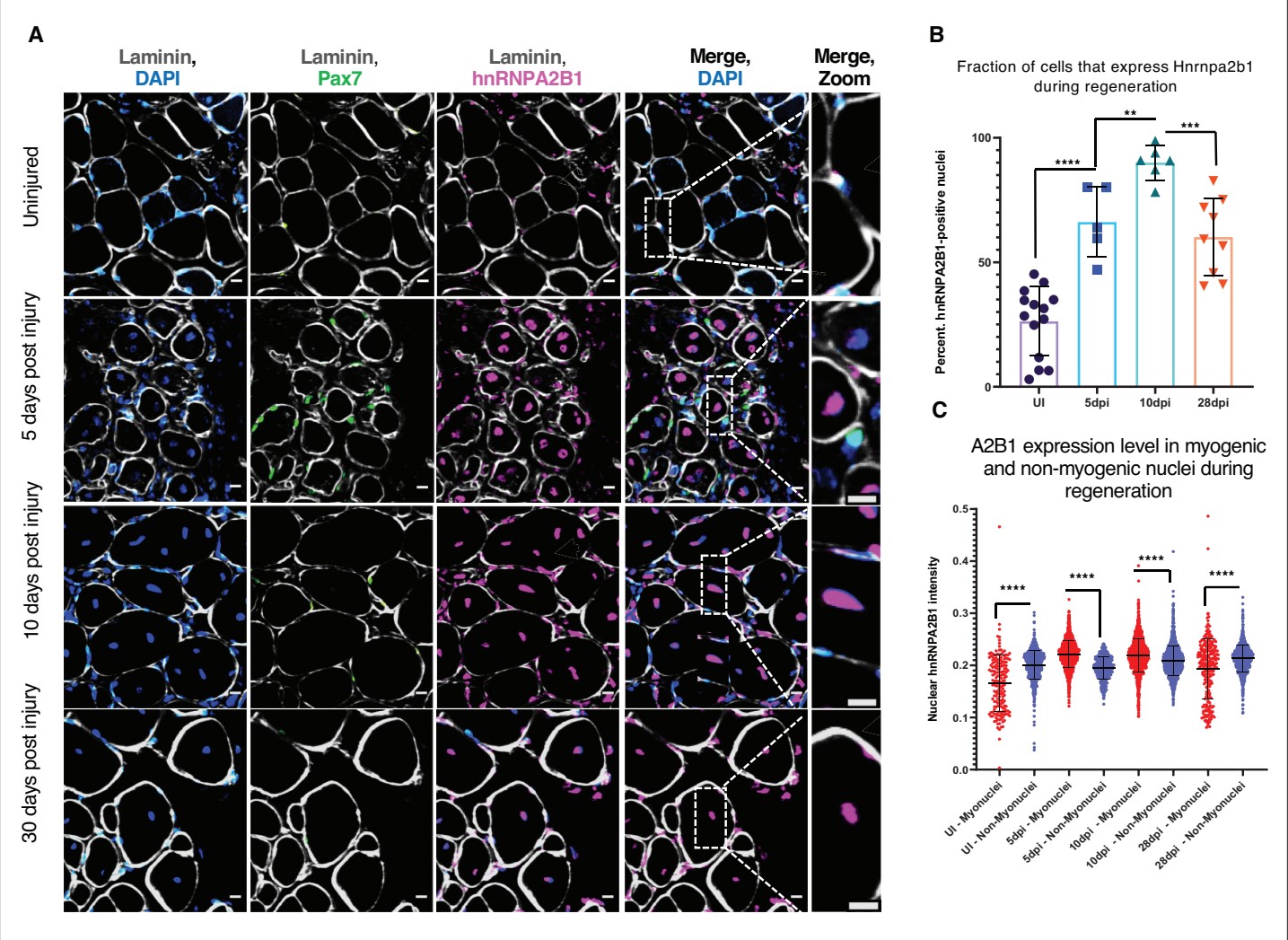

**Figure 3.** *Hnrnpa2b1* is upregulated in myogenic nuclei during skeletal muscle regeneration. (**A**) *Hnrnpa2b1* immunoreactivity in uninjured (UI), 5, 10, and 30 days post injury (dpi) regenerating mouse muscle. All images represent n = 3 biological replicates; scale = 20 μM. (**B**) Nuclear *Hnrnpa2b1* immunoreactivity in UI, 5, 10, and 28 dpi regenerating muscle. (**C**) Nuclear *Hnrnpa2b1* immunoreactivity intensity in either myonuclei or non-myonuclei in UI, 5, 10, and 28 dpi regenerating muscle. See also *Figure 3—figure supplement 1*.

The online version of this article includes the following source data and figure supplement(s) for figure 3:

**Figure supplement 1.** Related to *Figure 3*.

**Figure supplement 1—source data 1.** Related to *Figure 3B*.

*2018*) and proliferation (*He and Smith, 2009*). Thus, we hypothesized that *Hnrnpa2b1* may function as a myogenic RNA regulator during differentiation.

We examined the levels of Hnrnpa2b1 protein in regenerating muscle in vivo where in uninjured muscle, Hnrnpa2b1 is present in a subset of Pax7-positive MuSCs and in some peripherally located myonuclei in mature skeletal muscle fibers (*Figure 3A and B*, *Figure 3—figure supplement 1A*). By 5 dpi, Hnrnpa2b1 protein levels increase and Hnrnpa2b1 is present in the majority of Pax7-postitive MuSCs and centrally located myonuclei of immature regenerating myofibers (*Figure 3A–C*, *Figure 3—figure supplement 1A and B*). Hnrnpa2b1 levels peak at 10 dpi in both Pax7-positive MuSCs and regenerating myofibers and then Hnrnpa2b1 levels decline as myofibers fully regenerate by 28 dpi (*Figure 3A–C*, *Figure 3—figure supplement 1A,C and D*). Therefore, the changes in Hnrnpa2b1 protein levels correlate well with expression dynamics detected by single-cell sequencing in regenerating muscle.

These results show that Hnrnpa2b1 expression increases in both Pax7-positive MuSCs and regenerating myofibers early during regeneration . By contrast, *Tardbp* expression peaks at 5 dpi, near the peak of muscle progenitor proliferation (*Figure 3—figure supplement 1E–G*). As *Tardbp* is critical for early muscle regeneration, *Hnrnpa2b1* may play a similarly critical role later during muscle regeneration.

## *Hnrnpa2b1* is required for myoblast differentiation

To better understand the role of *Hnrnpa2b1* during myogenesis, we examined the protein levels of Hnrnpa2b1 in both proliferating and differentiating myoblasts. Hnrnpa2b1 immunoreactivity increases during myoblast differentiation, peaking by 3 days of differentiation, and declines as differentiating myoblasts and myotubes mature (*Figure 4A and B*). The levels of transcript mirror Hnrnpa2b1 expression in vivo and suggest a role for *Hnrnpa2b1* in regulating differentiation, and thus, we predict that a functional requirement for *Hnrnpa2b1* is highest during differentiation. To test this hypothesis, we knocked out *Hnrnpa2b1* in myoblasts using CRISPR/Cas9 and assessed the consequences of *Hnrnpa2b1* loss on myoblast proliferation and differentiation (*Figure 4C*, *Figure 4—figure supplement 1A and B*). *Hnrnpa2b1* wild type (WT) and knockout (KO) cells show no significant differences in proliferation when labeled with a timed pulse of the thymidine analog 5-ethynyl-2′-deoxyuridine (EdU; *Figure 4D and E*). The *Hnrnpa2b1* WT and KO populations exhibit no differences in nuclear morphology (*Figure 4—figure supplement 1E and F*).

Differentiation is impaired upon loss of *Hnrnpa2b1*. After 48 hr of differentiation, *Hnrnpa2b1* WT myoblasts had largely differentiated into multinucleated, myosin heavy chain-positive myotubes (*Figure 4F and G*). Conversely, *Hnrnpa2b1* KO myoblasts were unable to form large multinucleated myotubes (*Figure 4F and G*). The *Hnrnpa2b1* KO differentiation defect persisted after 3 days of differentiation in culture. These results suggest that *Hnrnpa2b1* KO myoblasts are unable to effectively differentiate (*Figure 4G*, *Figure 4—figure supplement 1C*). Thus, *Hnrnpa2b1* function is required for differentiation, but not proliferation.

## *Hnrnpa2b1* is a myogenic splicing regulator critical for terminal myogenic differentiation

We hypothesized that *Hnrnpa2b1* regulates RNA splicing during muscle differentiation, and that altered *Hnrnpa2b1*-mediated splicing may lead to impaired muscle differentiation. To test this hypothesis, we performed high-coverage RNA-sequencing of *Hnrnpa2b1* KO cells and WT differentiating myotubes. The transcripts detected correlate closely to previously published myogenic differentiation datasets, indicating negligible effects due to differing growth conditions (*Figure 5—figure supplement 2A and B*). Differential splicing analysis identifies 2167 alternatively spliced RNAs of which 40% are differentially expressed (*Figure 5A*, *Figure 5—figure supplement 2E*). Differential splicing analysis reveals that *Hnrnpa2b1* regulates the splicing of other RBPs. Many of these RBPs in turn regulate RNA splicing, including *Mbnl1*, *Mbnl2*, and *Rbfox2* (*Figure 5B*). Loss of *Rbfox2* or loss of both *Mbnl1* and *Mbnl2* impairs myogenic differentiation (*Lee et al., 2013*; *Singh et al., 2014*), and thus *Hnrnpa2b1* loss appears to lead to the exclusion of *Mbnl1*, *Mbnl2*, and *Rbfox2* exons encoding zinc fingers and RNA recognition motifs, respectively, which are predicted to disrupt function of these RBPs (*Figure 5B*). Thus, the splicing regulator cascade resulting from *Hnrnpa2b1* loss disrupts *Mbnl1*, *Mbnl2*, and *Rbfox2* splicing and may impair myogenesis.

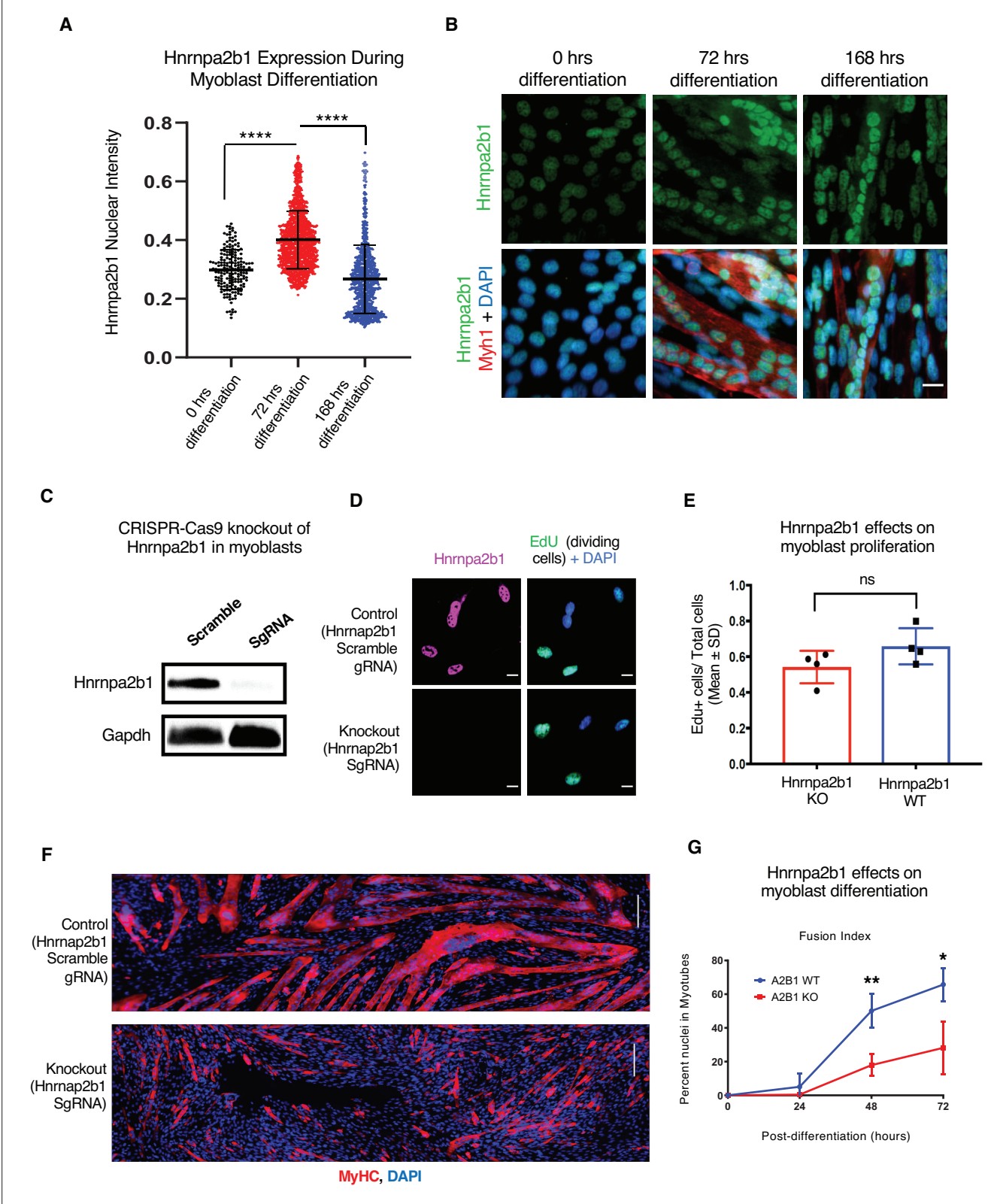

**Figure 4.** *Hnrnpa2b1* is required for myogenic differentiation. (**A**) *Hnrnpa2b1* nuclear protein in exponentially growing myoblasts (0 hr) and differentiating myotubes at 72 and 168 hr. (**B**) Immunoreactivity of *Hnrnpa2b1* in myoblasts and differentiating myotubes. (**C**) Western blot analysis of CRISPR-Cas9 knockout (KO) and scrambled sequence *Hnrnpa2b1* sgRNA-treated C2C12 myoblasts. (**D**) EdU-pulsed wild type (WT) myoblasts and *Hnrnpa2b1* KO myoblasts (scale = 10 μM). (**E**) Quantification of EdU incorporation in WT and KO *Hnrnpa2b1* myoblasts (ns = nonsignificant).

*Figure 4 continued on next page*

*Figure 4 continued*

(**F**) Immunoreactivity for *Myhc* in differentiating WT and KO *Hnrnpa2b1* myotubes (scale = 200 µM). (**G**) Quantification of Fusion Index (percentage of nuclei fused into myotubes) during differentiation in either WT or KO *Hnrnpa2b1* myotubes. All quantified data represent mean ± SD, two-tailed Student's *t*-test p-value: *<0.05, **<0.01, ***<0.001, ****<0.0001 unless otherwise stated. See also *Figure 4—figure supplement 1*.

The online version of this article includes the following source data and figure supplement(s) for figure 4:

**Source data 1.** Related to *Figure 4C*.

**Figure supplement 1.** Related to *Figure 4*.

We performed differential splicing analysis of *Mbnl1*, *Mbnl2*, and *Rbfox2* in differentiating *Hnrnpa2b1* KO cells (*Lee et al., 2013*; *Singh et al., 2014*). The alternatively spliced RNAs in *Hnrnpa2b1* KO cells, compared to *Mbnl1*, *Mbnl2* double KO cells, significantly overlap (hypergeometric p-value=1.2 × 10$^{-77}$). Similarly, alternatively spliced RNAs in *Hnrnpa2b1* KO cells and *Rbfox2* KO cells also overlap (hypergeometric p-value=6.1 × 10$^{-143}$) (*Figure 5C*), demonstrating that *Hnrnpa2b1* loss likely leads to altered splicing of RNAs regulated by *Mbnl1*, *Mbnl2*, and *Rbfox2,* respectively. As RBPs function together to co-regulate the splicing of target RNAs (*Klinck et al., 2014*; *Venables et al., 2013*), *Mbnl1*, *Mbnl2*, *Rbfox2*, and *Hnrnpa2b1* may cooperatively regulate target RNA splicing. Indeed, *Mbnl1*, *Mbnl2*, *Rbfox2*, and *Hnrnpa2b1* KOs exhibit identical splice site location and share changes in splicing of target RNAs (*Figure 5—figure supplement 1A*). Thus, a network of RBPs, including *Mbnl1*, *Mbnl2*, *Rbfox2*, and *Hnrnpa2b1,* appear to cooperatively regulate RNA splicing during muscle differentiation.

The RBPs cooperating in regulating splicing during muscle differentiation may function at different stages of differentiation, and thus, whether *Hnrnpa2b1* regulates the splicing of *Mbnl1*, *Mbnl2*, *Rbfox2*, and *Hnrnpa2b1* directly or whether the cooperation is in part due to sequencing cell populations at different stages of differentiation is unclear. To distinguish these possibilities, we mapped *Hnrnpa2b1* splicing at the single-cell level. To test whether *Hnrnpa2b1* loss results arrests myogenic cells at specific myogenic stages, we quantified the impact of *Hnrnpa2b1* KO on myogenic subcluster abundances. We employed CIBERSORTx, a machine learning tool, to impute myogenic subcluster percentages as CIBERSORTx is trained on single-cell sequencing data to estimate cell-type abundances present in bulk RNA-sequencing (*Newman et al., 2019*).

We first trained CIBERSORTx on our myogenic single-cell dataset and then imputed cell-type abundances in a muscle differentiation time course with bulk RNA sequences obtained from proliferating myoblasts and differentiating myotubes at 60 and 120 hr (*Trapnell et al., 2010*). Myogenic-trained CIBERSORTx reveals enrichment for proliferative MuSCs subclusters in the proliferating myoblast population. Conversely, the differentiating myotubes show a shift towards differentiating subclusters and are proportional to differentiation duration (*Figure 5—figure supplement 1B and C*). Myogenic-trained CIBERSORTx identifies a small fraction of differentiated cells within the proliferating myoblasts population (*Figure 5—figure supplement 1B and C*). To confirm whether the machine learning was accurate, we assessed immunoreactivity for myogenin, a marker of differentiation, in proliferating myoblasts and found myogenin-positive cells present at similar percentages as predicted by CIBERSORTx (*Figure 5—figure supplement 1C and D*). Together, these results demonstrate that myogenic-trained CIBERSORTx is capable of imputing myogenic subcluster abundances from bulk RNA-sequencing data.

Next, we tested the impact of *Hnrnpa2b1* KO using myogenic-trained CIBERSORTx. *Hnrnpa2b1* KO cells show an enrichment for subcluster 5 myogenic cell signatures and a decrease in subcluster 7 (*Figure 5D*), demonstrating that *Hnrnpa2b1* KO cells are slow to transition to subcluster 7 and accumulate in subcluster 5. We then used CIBERSORTx to examine the impact of *Hnrnpa2b1* KO on specific myogenic cell states. We first performed differential gene expression analysis for each of the subclusters identified by CIBERSORTx (clusters 2–5, 7) to identify differentially expressed genes enriched in the cells of each specific cluster. These differentially expressed genes provide a molecular signature for the cells at each of these specific stages embedded in bulk RNA-sequencing data. By examining the splicing changes for the differentially expressed genes, we found that *Hnrnpa2b1* KO alters splicing in each of these clusters, with cluster 5 showing the most changes in splicing (*Figure 5E*, *Figure 5—figure supplement 1E*, *Figure 5—figure supplement 2F*). Using this approach, we can

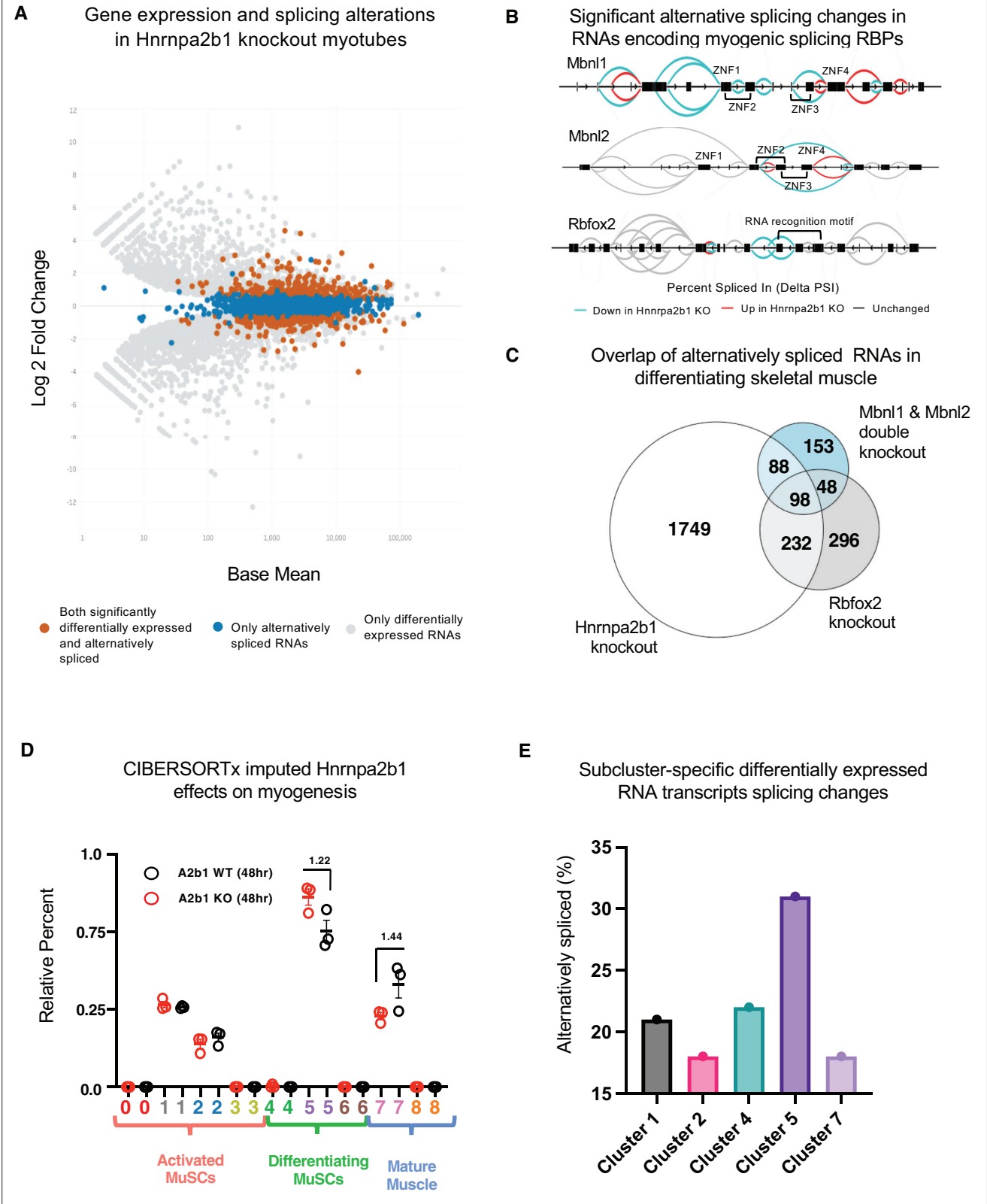

**Figure 5.** *Hnrnpa2b1* is a myogenic splicing regulator critical for terminal myogenic differentiation. (**A**) Differential gene expression identified by DESeq2 in wild type (WT) and *Hnrnpa2b1* knockout (KO) myotubes after 48 hr of differentiation with alternative splicing changes identified by LeafCutter (differential gene expression significance p-adjusted<0.05, splicing false discovery rate [FDR] p-value<0.05). (**B**) Sashimi plots of significant alternative splicing changes (delta percent spliced in [dPSI]) for *Mbnl1, Mbnl2, Rbfox2* in differentiating *Hnrnpa2b1* KO myotubes. (**C**) Venn diagram of significantly

*Figure 5 continued on next page*

*Figure 5 continued*

altered spliced transcripts in *Hnrnpa2b1* KO, *Mbnl1/Mbnl2* double KO (*Thomas et al., 2017*), and *Rbfox2* KO (*Singh et al., 2014*) differentiating myogenic cultures. (**D**) Myogenic-trained CIBERSORTx-imputed stacked bar chart of myogenic subcluster percentages in C2C12 differentiation time course (*Trapnell et al., 2010*). (**E**) Myogenic-trained CIBERSORTx machine learning imputed percentages of myogenic clusters from WT and KO *Hnrnpa2b1* differentiating myotubes. Y-axis refers to fold change between myogenic cluster percentages. See also *Figure 5—figure supplement 1*.

The online version of this article includes the following figure supplement(s) for figure 5:

**Figure supplement 1.** Related to *Figure 5*.

**Figure supplement 2.** Related to *Figure 5* – a distinct *Hnrnpa2b1* gene expression and splicing program during differentiation.

control for population-wide cell state differences and demonstrate that *Hnrnpa2b1* is a key splicing regulator in specific differentiating myogenic cell states.

## RBP engagement scoring delineates timing of RBP function during myogenesis

The expression of an RBP and RBP target RNAs is necessary for an RBP to alter splicing of the RPB target. Therefore, expression of *Hnrnpa2b1* target RNAs during myogenic differentiation governs when *Hnrnpa2b1* is capable of splicing target transcripts. To identify *Hnrnpa2b1* target RNAs during myogenesis, we performed *Hnrnpa2b1* enhanced UV crosslinking and immunoprecipitation (eCLIP) in both myoblasts and myotubes (*Figure 6A*, *Figure 6—figure supplement 1A–C*). *Hnrnpa2b1* eCLIP reveals 808 binding sites across 247 genes for myoblasts and 1030 binding sites across 137 genes for myotubes, which are significantly enriched compared to size-matched input (reflecting all RNA–protein interactions in the input) (*Figure 6—figure supplement 1D*). *Hnrnpa2b1* RNA target eCLIP peaks were highly correlated between biological replicates, had thousands of reproducible eCLIP clusters by irreproducible discovery rate analysis, and capture prior identified *Hnrnpa2b1* RNA targets including *Hnrnpa2b1's* own 3′UTR (*Figure 6—figure supplement 1D–G*; *Martinez et al., 2016*). Analysis of *Hnrnpa2b1* RNA-binding sites reveals an enrichment for binding sites in the 3′UTR and proximal introns of target RNAs, potentially indicating a role of *Hnrnpa2b1* in RNA localization or translation (*Figure 6A*). Further, *Hnrnpa2b1* binding maps closely to splicing clusters, indicating a role for *Hnrnpa2b1* in splicing regulation (*Figure 6—figure supplement 2B and C*). *Hnrnpa2b1* binding spatially correlates to splicing changes in the two significantly differentially spliced transcripts *Hnrnpa2b1* and *Myl1* (*Figure 6—figure supplement 3A and B*). We validated the interaction of *Hnrnpa2b1* with several target RNAs using RIP qRT-PCR (*Figure 6B*, *Figure 6—figure supplement 1H*).

Having defined *Hnrnpa2b1* mRNA targets, we next looked to develop a computational toolset to identify myogenic cells in which *Hnrnpa2b1* is functioning. We reasoned that many RBPs engage with target RNAs co-transcriptionally to regulate RNA splicing (*Fu and Ares, 2014*). Newly transcribed RNAs or pre-mRNAs can be identified as RNAs that contain introns. These newly transcribed pre-mRNAs can then be computationally defined in single-cell data (*La Manno et al., 2018*) where RBP function will correlate with the expression level of an RBP and the RBP's target pre-mRNA. For example, RBP function is predicted to occur in cells with both high RBP expression and target pre-mRNAs expression, which we term 'RBP engagement scoring' (*Figure 6C*).

We validated RBP engagement scoring on a well-characterized RBP, *Tardbp*, a splicing RBP required for myoblast proliferation. Prior eCLIP experiments identified *Tardbp* RNA targets (*Vogler et al., 2018*), and if our computation approach is valid, *Tardbp* engagement scores should be higher in proliferating myoblasts than differentiated myotubes. We quantified *Tardbp* and target RNA expression in single cells, assigned engagement scores to each of these cells, and then examined the engagement scores for all the cells in a given subcluster to examine the impact of *Tardbp* at each stage of myogenesis. *Tardbp* engagement scores are higher in proliferating myoblasts than in myogenic cells undergoing differentiation (*Figure 6D*), consistent with the functional requirement of *Tardbp* in proliferating myoblasts. We then performed RBP engagement scoring for *Hnrnpa2b1* by examining the pre-mRNA levels of *Hnrnpa2b1* target RNAs and *Hnrnpa2b1* in single cells and correlated the expression of *Hnrnpa2b1* target RNAs to *Hnrnpa2b1* expression. Myogenic subclusters 5 and 6 showed the highest *Hnrnpa2b1* engagement scores (*Figure 6D*), suggesting that *Hnrnpa2b1* splicing function is highest in

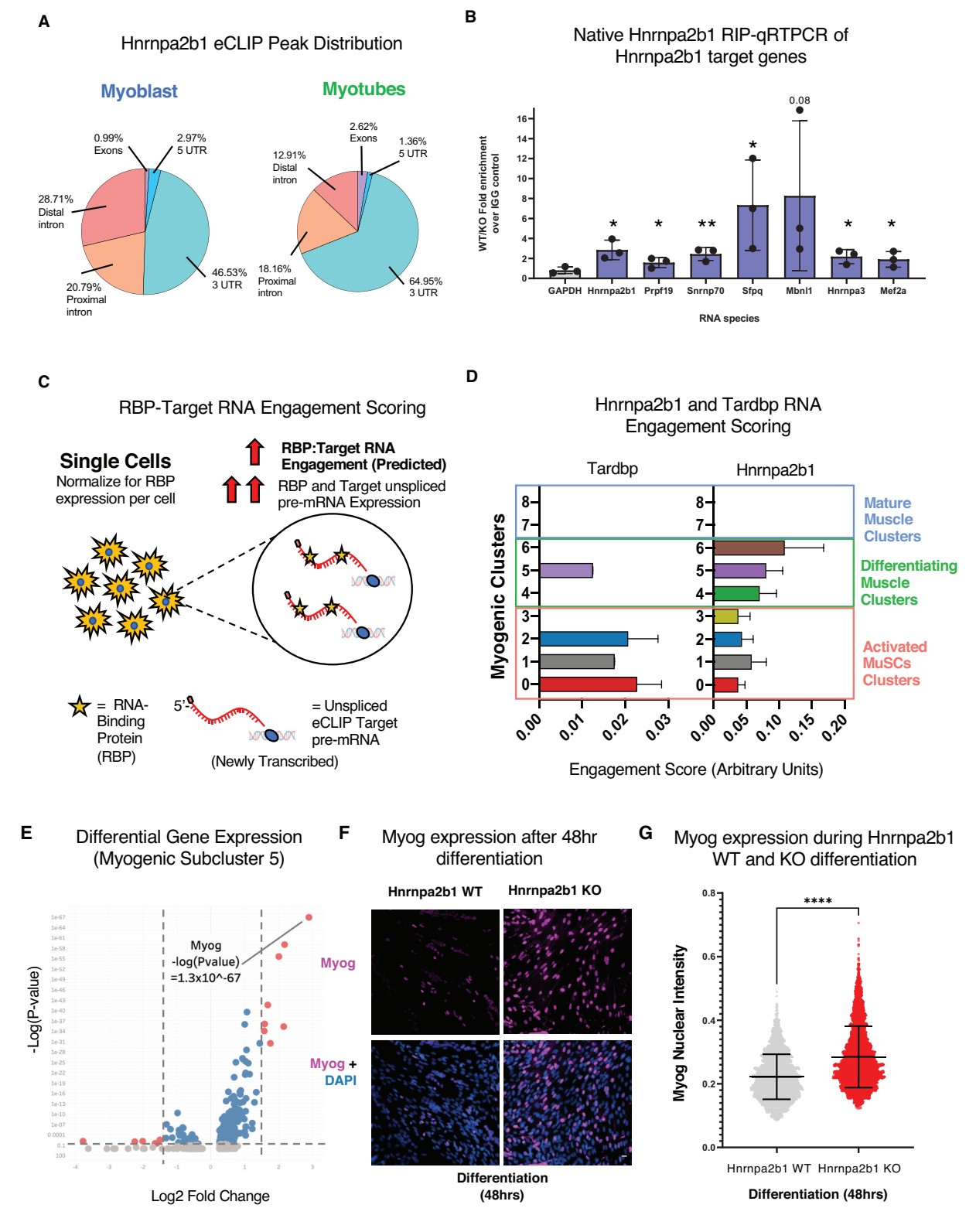

**Figure 6.** RNA engagement scoring delineates functional timing of RNA-binding proteins (RBPs) during myogenesis. (**A**) Genomic distribution of *Hnrnpa2b1* enhanced UV crosslinking and immunoprecipitation (eCLIP) peaks in myoblasts and myotubes. (**B**) Native (non-denaturing) *Hnrnpa2b1* RNA immunoprecipitation (RIP)-qRT-PCR of RNA targets identified by eCLIP. (**C**) Schematic for RBP-Target RNA engagement scoring. (**D**) RBP-Target RNA engagement scoring of *Tardbp* and *Hnrnpa2b1* in myogenic subclusters (median ± 95% CI across all RBP-RNA engagement scores in each myogenic

*Figure 6 continued on next page*

*Figure 6 continued*

subcluster). (**E**) Differential gene expression analysis of myogenic subcluster 5. (**F**) *Myog* immunoreactivity in wild type (WT) and *Hnrnpa2b1* knockout (KO) 48 hr differentiated myotubes. (**G**) *Myog* immunoreactivity signal intensity in differentiating WT and *Hnrnpa2b1* KO myotubes (mean ± SD, two-tailed Student's *t*-test p-value: ****<0.0001). See also *Figure 6—figure supplement 1*.

The online version of this article includes the following source data and figure supplement(s) for figure 6:

**Figure supplement 1.** Related to *Figure 6*.

**Figure supplement 1—source data 1.** Related to *Figure 6—figure supplement 1A-C*.

**Figure supplement 2.** Related to *Figure 6* – magnitude and spatial location for significant Hnrnap2b1 knockout (KO) splicing alterations.

**Figure supplement 3.** Related to *Figure 6* – mapped relationship of enhanced UV crosslinking and immunoprecipition (eCLIP) peaks to splicing change magnitude in significantly differentially spliced transcripts.

cells in subclusters 5 and 6. Thus, an *Hnrnpa2b1* KO may arrest cells at these stages, causing impaired differentiation.

Delayed or slowed progression through subcluster 6 would lead to cells accumulating in the preceding subcluster 5. To test if *Hnrnpa2b1* KO leads to an accumulation of cells in subcluster 5, we performed differential gene expression analysis on myogenic subclusters 5 and 6. *Myog* is significantly enriched in subcluster 5 (p-value=$1 \times 10^{-67}$) (*Figure 6E*, *Figure 6—figure supplement 1I*). We examined myogenin immunoreactivity in differentiated *Hnrnpa2b1* KO and WT cells. *Hnrnpa2b1* KO cells are significantly enriched in the myogenin-positive cell population (*Figure 6F and G*), demonstrating a requirement for *Hnrnpa2b1* for commitment to terminal differentiation.

## Discussion

RBP dysfunction contributes to neuromuscular disease (*Apponi et al., 2011*; *Farina et al., 2012*), yet the mechanisms leading to disease phenotypes and the roles of RBPs in regulating myogenic cell fate decisions remain poorly understood. Using multiomic and single-cell technologies, we finely mapped RBP expression during myogenesis, discovering that RBP expression and function correlates with specific myogenic cell states. We define a role for the disease-associated RBP, *Hnrnpa2b1*, in directing MuSC commitment to terminal differentiation where *Hnrnpa2b1* influences global RNA splicing by regulating RBPs that splice distinct target mRNAs. The computational tools we employed allowed us to pinpoint the precise cell state in which *Hnrnpa2b1* functions to guide cell fate decisions.

We propose a model to temporally define RBP function during muscle regeneration whereby the expression of an RBP and the RBP's target RNAs is required, providing precise timing for RPB function (*Figure 7*). Whether an RBP is functioning will depend upon the expression of the RBP as well as expression of the RBP target RNAs. Thus, even though an RBP transcript is present, the RBP will not function unless the target RNAs are present, ensuring that RBPs do not function ubiquitously during muscle repair or muscle development. The observation that one splicing RBP regulates the splicing of an RBP that in turn regulates splicing of RNA targets adds combinatorial control and redundancy to myogenic differentiation. The network of shared RBPs ensures that splicing and RNA regulation function during a specific cell state, which we propose is a universal property occurring during development and in a number of regenerating tissues and organs.

The model we propose explains the seemingly discordant myogenic phenotypes resulting from mutations in different RBPs that are consistent with the observation that KOs of distinct RBPs exhibit overlapping phenotypes . We speculate that RBPs function in concert at a specific cell state to regulate a wide array of messenger RNA targets that coordinate cell fate transitions. RBP dysfunction in diseases is therefore likely cell state-defined and RBP mutations disproportionately impact a specific cell's state. The resultant effect of an RPB mutation is delayed cell fate transition, a complete block of cell fate transition, or selection of an alternative cell fate. A better understanding of the complex events contributing to alterations in cell state transitions occurring when RBPs are mutated could lead to the development of new approaches for cell-state-specific therapeutic intervention (*Ferlini et al., 2021*).

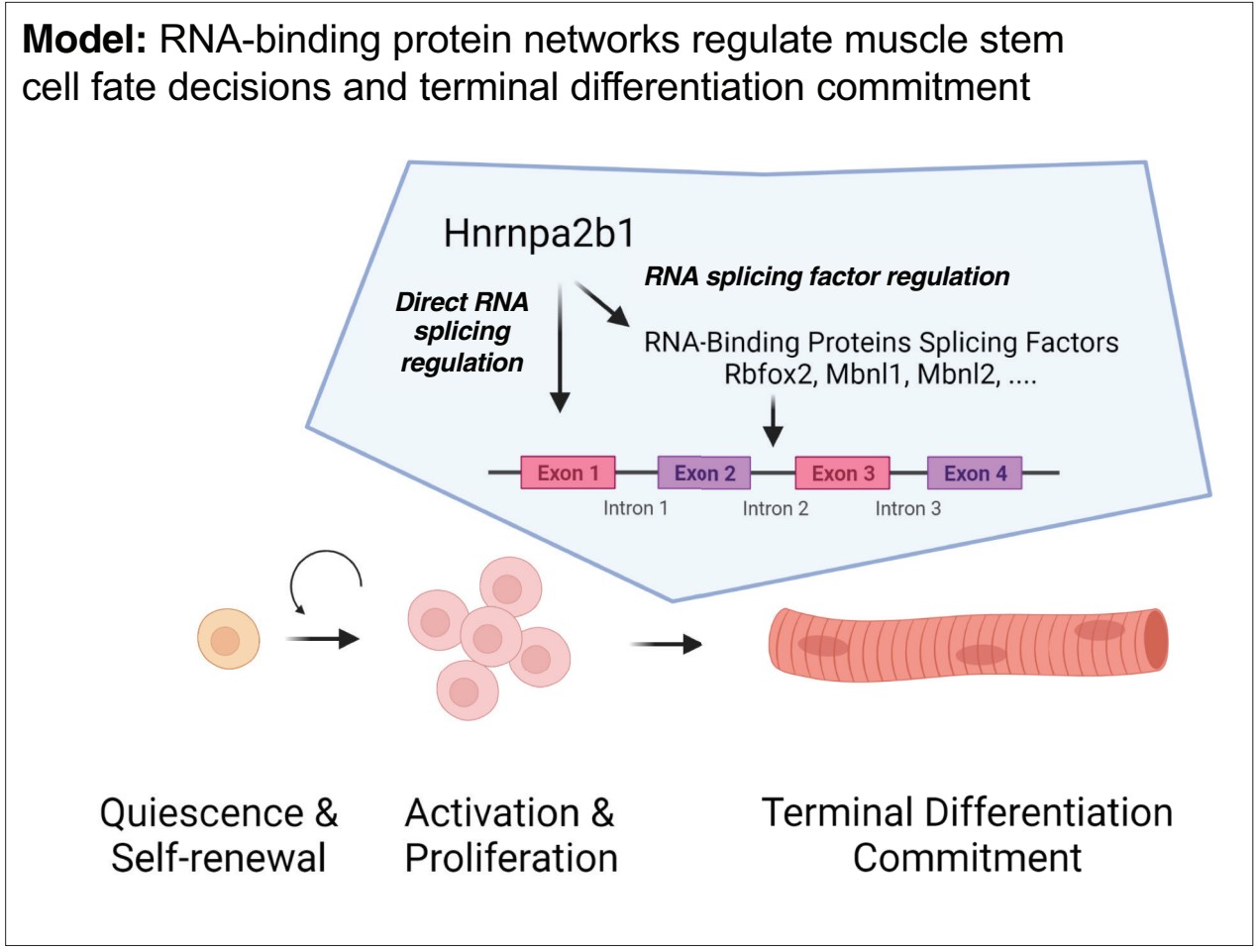

**Figure 7.** Discrete RNA-binding protein functional timing and expression as a dynamic post-transcriptional regulatory mechanism for directing myogenic fate decisions.

## Materials and methods

### Mice

Mice were bred and housed according to the National Institutes of Health (NIH) guidelines for the ethical treatment of animals in a pathogen-free facility at the University of Colorado at Boulder. The University of Colorado Institutional Animal Care and Use Committee (IACUC) approved all animal protocols and procedures and studies complied with all ethical regulations. IACUC protocol number 2516, Animal Welfare Assurance number A3646-01. WT mice were genotype C57BL/6 (Jackson Laboratories). Cells and TA muscles were isolated from 3- to 6-month-old male and female WT mice.

### Mouse injuries

Male and female mice between 3 and 6 months of age were anesthetized with isoflurane and the left TA muscle was injected with 50 µl of 1.2% BaCl$_2$. The injured and contralateral TA muscles were harvested at the indicated time points.

### TA collections and cell isolations

TA muscles were dissected and placed into 400 U/ml collagenase at 37°C for 1 hr with shaking and then placed into Ham's F-12C supplemented with 15% horse serum to inactivate the collagenase. Cells were passed through three strainers of 100, 70, and 40 µm (BD Falcon) and flow through was centrifuged at 1500 × g for 5 min, and the cell pellets were resuspended in Ham's F-12C. To remove dead cells and debris, cells were passed over Miltenyi dead cell removal kit columns (Cat# 130-090-101). To remove RBCs, cells were incubated with antiTer119 micro magnetic beads and passed over a Miltenyi

column (Cat# 130-049-901). For adult uninjured Tas, six TA muscles (from three mice) were pooled together. For injured TA muscles, two TA muscles from two different mice were pooled together. Cells were then counted using a Bio-Rad TC20 automated cell counter and processed with a 10X Genomics single-cell sequencing kit.

## Single-cell sequencing

To capture, label, and generate transcriptome libraries of individual cells, we used the 10X Genomics Chromium Single Cell 3′ Library and Gel Bead Kit v2 (Cat# PN-120237) following the manufacturer's protocols. Briefly, the single-cell suspension, RT-PCR master mix, gel beads, and partitioning oil were loaded into a Single Cell A Chip 10 genomics chip, placed into the chromium controller, and the Chromium Single Cell A program was run to generate Gel Bead-In-EMulsion (GEMs) that contain RT-PCR enzymes, cell lysates and primers for Illumina sequencing, barcoding, and poly-DT sequences. GEMs were then transferred to PCR tubes and the RT-PCR reaction was run to generate barcoded single-cell identified cDNA. Barcoded cDNA was used to make sequencing libraries for analysis with Illumina sequencing. We captured 1709 cells from young uninjured muscle, 5077 from the 4 dpi muscle, and 2668 from the 7 dpi muscle. Sequencing was completed on an Illumina NovaSeq 6000 using paired-end 150 cycle 2 × 150 reads by the genomics and microarray core at the University of Colorado Anschutz Medical Campus.

## Single-cell Informatics

Preprocessing was performed using Cellranger v3.0.1 (10X Genomics) count module that was used for alignment using cellranger-mm10-3.0.0 refdata, filtering, barcode counting, and UMI counting of the single-cell FASTQs. Postprocessing was performed using Seurat according to standardized workflows (*Butler et al., 2018*; *Stuart et al., 2019*). In brief, using RStudio (v4.0.0), Seurat objects were created for each Cellranger processed sample by importing filtered_gene_bc_matrices. Multiple Seurat objects then were merged, filtered, normalized, feature selected, scaled, and clustered. Nonlinear dimensional reduction was performed using uniform manifold approximation and projection (UMAP).

*RNA velocity* using the scVelo pipeline was performed according to standardized workflows (*La Manno et al., 2018*). In brief, Velocyto was run in Python v3.6.3 using Samtools v1.8, cellranger-mm10-3.0.0 refdata, and masking repetitive elements to generate unspliced/spliced/ambiguous read count Loom file for each 10X cellranger preprocessed library. Seurat objects of myogenic clusters (.rds files) were converted in RStudio to loom files using LoomR. In a virtual environment, Velocyto Loom files were concatenated and merged with Seurat exported myogenic Loom in Scanpy. The scVelo pipeline was then used to perform log normalization and filtering. RNA velocity was then imputed using stochastic modeling. RNA velocities were projected on pre-computed UMAP embeddings and annotated clusters.

*RBP Engagement Scoring* was performed by first determining the raw unspliced reads for every gene at single-cell level using Veloctyo as above. Metadata containing raw reads counts and unspliced reads imputed using Veloctyo were exported from Scanpy anndata objects for single cells contained within myogenic clusters and were exported and reimported into RStudio (v.4.0.0). To calculate an RBP-RNA engagement score for a given RBP, CLIP target genes of a given RBP genes were subset and unspliced read counts and total RBP counts were normalized to total read counts in a given cell. Scoring then represents the fraction of normalized unspliced reads of target RNAs to normalized amounts of a given RBP. To extrapolate to a given cluster, scores were then summed across all cells in a given cluster and data was represented by the median of these scores for a given cluster of cells. Target genes were identified using publicly available eCLIP datasets for *Tardbp*.

*CIBERSORTx* of myogenic clusters was performed according to published workflows (*Newman et al., 2019*). In brief, a single-cell reference txt file was created from our Seurat processed single-cell RNA-seq dataset. A mixture file for bulk RNA-sequencing samples was created using TPM values extracted from StringTie serve as input. A single-cell reference txt matrix of cells organized by myogenic clusters was then used to train CIBERSORTx using default parameters (*Supplementary file 3*). Cell fractions were then imputed using 100 permutations for significance analysis.

## Immunofluorescence staining of tissue sections

TA muscles were dissected, fixed on ice for 2 hr with 4% paraformaldehyde, and then transferred to PBS with 30% sucrose at 4°C overnight. Muscle was mounted in O.C.T. (Tissue-Tek) and cryo-sectioned on a Leica cryostat to generate 10-μm-thick sections. Frozen tissues and sections were stored at –80°C. Prior to immunofluorescent staining, tissue sections were post-fixed in 4% paraformaldehyde for 10 min at room temperature (RT) and washed three times for 5 min in PBS. Staining with anti-Pax7, Laminin, and Hnrnpa2b1 antibodies required heat-induced epitope retrieval, which was performed by placing post-fixed slides in citrate buffer, pH 6.0, and subjected them to 6 min of high-pressure cooking in a Cuisinart model CPC-600 pressure cooker. For immunostaining, tissue sections were permeabilized with 0.25% Triton-X100 (Sigma) in PBS containing 2% bovine serum albumin (Sigma) for 60 min at RT. Incubation with primary antibody occurred at 4°C overnight followed by incubation with secondary antibody at RT for 1 hr. Primary antibodies included anti-Pax7 (Developmental Studies Hybridoma Bank, University of Iowa, USA) at 1:750, rabbit anti-laminin (Sigma-Aldrich) at 1:200, and a mouse anti-Hnrnpa2b1 (ab6102, Abcam) at 1:200. Alexa Fluor secondary antibodies (Molecular Probes) were used at a 1:1000 dilution. For analysis that included EdU detection, EdU staining was completed prior to antibody staining using the Click-iT EdU Alexa Fluor 488 detection kit (Molecular Probes) following the manufacturer's protocols. Sections were incubated with 1 μg/ml DAPI for 10 min at RT then mounted in Mowiol supplemented with DABCO (Sigma-Aldrich) or ProLong Gold (Thermo) as an anti-fade agent. All microscopy images used for quantitation were taken of samples cultured, immunohistochemically stained, and imaged in parallel and under identical conditions to enable quantitative comparison.

## Cell culture and growth conditions

### C2C12 myoblast cells

Immortalized murine myoblasts (American Type Culture Collection CRL-1772) were maintained on uncoated standard tissue culture plastic or gelatin-coated coverslips for imaging experiments at 37°C with 5% $CO_2$ in Dulbecco's Modified Eagle Medium (DMEM) with 20% fetal bovine serum and 1% penicillin/streptomycin. To induce myoblast differentiation and fusion into myotubes, C2C12 myoblasts at 80% confluence were switched to DMEM media supplemented with 5% horse serum, 1% penicillin/streptomycin, and 1X Insulin-Transferrin-Selenium in DMEM. Cell lines were validated using RNA deep sequencing concurrent with certification provided by the manufacturer. Cells tested negative for mycoplasma contamination by the BioFrontiers cell culture core facility.

### *Hnrnpa2b1* CRISPR-Cas9 knockout and EdU incorporation

CRISPR-Cas9 knockout was performed in C2C12 myoblasts. Single-guide RNA (sgRNA) against *Hnrnpa2b1* (5'-GAGTCCGCGATGGAG) were designed using (crispr.mit.edu) and cloned into pSpCas9(BB)–2A-Puro (PX459). C2C12 myoblasts were transfected with JetPrime using the manufacturer's protocols. Myoblasts that integrated the CRISPR construct were selected with puromycin (1 μg/ml) for 1 week. Three independent myoblasts KO and WT clones were isolated using a cloning ring to selectively detach clonal populations via trypsinization. Clonal population KO was validated via immunofluorescence (IF) and Western blotting against *Hnrnpa2b1* with anti-Hnrnpa2b1 antibodies (ab6102 and ab31645). *EdU incorporation*: C2C12 myoblasts were incubated with 10 μM EdU (Life Technologies) for 3 hr. Cells were washed, fixed, and stained using the methods described below.

## Immunofluorescence staining of cells

C2C12 myoblast cells were washed with PBS in a laminar flow hood and fixed with 4% paraformaldehyde for 10 min at RT in a chemical hood. Cells were permeabilized with 0.25% Triton-X100 in PBS containing 2% bovine serum albumin (Sigma) for 1 hr at RT. Cells were incubated with primary antibody at 4°C overnight, then incubated with secondary antibody at RT for 1 hr. Primary antibodies included mouse anti-Hnrnpa2b1 (ab6102, Abcam) at 1:200, mouse-anti-myogenin (ab82843, Abcam), and a mouse anti-MHC MF-20 (Developmental Studies Hybridoma Bank, University of Iowa) at undiluted, 'neat' concentration. Alexa Fluor secondary antibodies (Molecular Probes) were used at a 1:1,000 dilution.

## Microscopy and image analyses

Images were captured on a Nikon inverted spinning disk confocal microscope. Objectives used on the Nikon were ×10/0.45 NA Plan Apo, ×20/0.75 NA Plan Apo, and ×40/0.95 Plan Apo. Confocal

stacks were projected as maximum intensity images for each channel and merged into a single image. Brightness and contrast were adjusted for the entire image as necessary against secondary antibody-treated control immunofluorescent sections. Cellprofiler was used to quantify immunohistochemistry (IHC) and IF images using custom analysis pipelines unless otherwise stated.

## Western blotting of cell and tissue lysates

Western blot was performed according to standard protocols. Equal volumes (20 µl) of fractions were then resolved on a 4–12% Bis-Tris SDS-PAGE gel and transferred to a nitrocellulose membrane (Bio-Rad). Primary antibodies included mouse anti-Hnrnpa2b1 (1:200; ab6102, Abcam) and monoclonal rabbit anti-GAPDH (14C10) conjugated to HRP (1:2000; Cell Signaling, 3683S).

## *Hnrnpa2b1* eCLIP sequencing

C2C12 myoblasts were seeded at $6 \times 10^6$ cells per 15 cm plate, grown 24 hr at 37°C, 5% $CO_2$, and either harvested (undifferentiated myoblasts) or differentiated in differentiation media for 4 days. Hnrnpa2b1 eCLIP was performed according to established protocols (*Nguyen et al., 2018*; *Van Nostrand et al., 2016*). In brief, *Hnrnpa2b1*-RNA interactions were stabilized with UV crosslinking (254 nm, 150 mJ/$cm^2$). Cell pellets were collected and snap-frozen in liquid $N_2$. Cells were thawed, lysed in eCLIP lysis buffer (50 mM Tris-HCl pH 7.4, 100 mM NaCl, 1% NP-40, 0.1% SDS, 0.5% sodium deoxycholate, and 1× protease inhibitor), and sonicated (Bioruptor). Lysate was RNAse I (Ambion, 1:25) treated to fragment RNA. Protein-RNA complexes were immunoprecipitated using the rabbit polyclonal anti-Hnrnpa2b1 (ab31645, Abcam) antibody. One size-matched input (SMInput) library was generated per biological replicate using an identical procedure without immunoprecipitation. Stringent washes were performed as described, RNA was dephosphorylated (FastAP, Fermentas), T4 PNK (NEB), and a 3′ end RNA adaptor was ligated with T4 RNA ligase (NEB). Protein-RNA complexes were resolved on an SDS-PAGE gel, transferred to nitrocellulose membranes, and RNA was extracted from membrane. After RNA precipitation, RNA was reverse-transcribed using SuperScript IV (Thermo Fisher Scientific), free primer was removed, and a 3′ DNA adapter was ligated onto cDNA products with T4 RNA ligase (NEB). Libraries were PCR-amplified and dual-indexed (Illumina TruSeq HT). Pair-end sequencing was performed on Illumina NextSeq sequencer.

## eCLIP bioinformatic and statistical analysis

Read processing and cluster analysis for *Hnrnpa2b1* eCLIP were performed as previously described (*Van Nostrand et al., 2016*; *Vogler et al., 2018*). Briefly, 3′ barcodes and adapter sequences were removed using standard eCLIP scripts. Reads were trimmed, filtered for repetitive elements, and aligned to the mm9 reference sequence using STAR. PCR duplicate reads were removed based on the read start positions and randomer sequence. Bigwig files for genome browser display were generated based on the location of the second paired-end reads. Peaks were identified using the encode_branch version of CLIPPER using the parameter '-s mm9.' Peaks were normalized against size-matched input by calculating fold enrichment of reads in IP versus input and were designated as significant if the number of reads in the IP sample was greater than in the input sample, with a Bonferroni corrected Fisher's exact p-value$<10^{-8}$.

## RNA-sequencing library preparation and sequencing

C2C12 *Hnrnpa2b1*-WT and KO myoblasts were seeded at $6 \times 10^6$ cells per 15 cm plate, grown for 24 hr at 37°C, 5% $CO_2$, and differentiated via serum withdrawal and ITS supplementation for 2 days. Differentiated C2C12 myoblasts and myotubes were washed with PBS in a laminar flow hood and scraped from the tissue culture plate. Total RNA was extracted using a QIAGEN RNeasy Plus Mini Kit, following the manufacturer's instructions. Isolated RNA quality was assessed by the CU Boulder BioFrontiers Next-Gen Sequencing core facility using an Agilent tape station. Isolated RNA was sent to the University of Colorado Cancer Center Genomics and Microarray Core for 'Ribominus' Ribosomal RNA depletion, NGS library preparation, and total RNA sequencing.

## RNA-sequencing informatics

All RNA-sequencing data preprocessing was carried using standardized Nextflow RNA-sequencing (nf-core/rnaseq) with STAR alignment (genome GRCm39), transcriptome mapping with Salmon, and in

silico ribosomal depletion using SortMeRNA. Differential gene expression was performed on feature counts tables using DESeq2 (*Love et al., 2014*). RNA splicing analysis was performed using Leaf-Cutter (*Li et al., 2018*).

## CLIP/splicing cluster analysis

The locations of splicing clusters generated using LeafCutter and *Hnrnpa2b1* eCLIP peaks from myotubes were each compared against genes in the mm10 refGene table from the UCSC Genome Browser to determine the number of clusters or peaks overlapping each gene. The overlap of each cluster or peak was then subdivided into coverage of either UTR or segments of the gene, divided into percentiles along the gene. These were normalized and plotted against each other using R and ggplot2.

## Relative distance

Relative distance between splicing clusters generated using LeafCutter and *Hnrnpa2b1* eCLIP peaks from myotubes were generated using the bedtools reldist command, using eCLIP peaks as the first input, and splicing clusters in genes with eCLIP peaks as the second input (*Favorov et al., 2012*).

## QAPA poly-A analysis

As previously described, RNA-sequencing reads were trimmed, in silico ribodepleted, and reads were mapped to an mm10 3′UTR annotation file using Salmon v0.13.1. Quantification of alternative polyadenylation (QAPA) was then used to estimate relative alternative 3′UTR isoform usage (*Ha et al., 2018*). Lengthening or shortening of a gene's poly(A) tail was determined by first calculating a proximal poly(A) site usage (PPAU) defined as the percentage of reads mapping to the most proximal poly(A) site relative to reads mapping to the whole 3′UTR. ΔPPAU was then calculated as the median PPAU *Hnrnpa2b1* KO – median PPAU WT (three replicates per condition). A gene with a ΔPPAU >20 was defined as having a shortened poly(A) tail, and a gene with a ΔPPAU < –20 was defined as having a lengthened poly(A) tail.

## *Hnrnpa2b1* immunoprecipitation and qRT-PCR

C2C12 *Hnrnpa2b1*-WT and KO myoblasts were seeded at $6 \times 10^6$ cells per 15 cm plate, grown 24 hr at 37°C, 5% $CO_2$, and harvested as undifferentiated myoblasts. Myoblasts were lysed in a CHAPS-based buffer and pre-cleared using protein-A bound Dynabeads (Thermo, 10001D). Pre-cleared cell lysates were incubated with rabbit polyclonal anti-Hnrnpa2b1 antibody (ab31645, Abcam). Antibody-bound *Hnrnpa2b1* was bound to protein-A Dynabeads overnight and magnetically isolated from the whole-cell lysate. *Hnrnpa2b1*-bound RNA was isolated from the Dynabead-Antibody-Hnrnpa2b1 complex via TRIZol RNA purification. cDNA libraries were created from purified RNA via oligo-DT priming and SuperScriptIII-enzyme reverse transcription. cDNA libraries were probed against *Hnrnpa2b1*-eCLIP hits chosen as a subset of the significantly enriched Gene Ontology terms via quantitative, real-time PCR (qRT-PCR). Primers used target *Gapdh, Hnrnpa2b1, Prpf19, Snrnp70, Sfpq, Mbnl1, Hnrnpa3,* and *Mef2a*. (Primer sequence is given in *Supplementary file 1*.) qRT-PCR was performed using SYBR-Green qRT-PCR reagent (Bio-Rad), and fluorescent emission was measured using a Bio-Rad CFX384 Real-Time PCR Detection system.

## Acknowledgements

We thank J Dragavon, J Orth for help with microscopy and the CU Microscopy Cores (supported by NIST-CU 70NANB15H226, NIH1S10RR026680-01A1), as well as the University of Colorado Cancer Center Genomics Core (supported by NIH-P30CA46934). We thank Theodore E Ewachiw and Yang Zhao for their technical assistance. The research was supported by NIH-T32GM008497 (JRW, EDN, TOV, and EL), NIH-F30NS093682 (JRW), NIH-F30AR068881 (TOV), NIH-GM045443 (RP), NIH-R35GM119575 (AMJ), Paul O'Hara II Seed Grant from ACS-IRG Grant Program (AMJ), NIH-AR049446 and NIH-AR070360 (BBO), NIH-RM1-HG007735 (HYC), Glenn Foundation for Biomedical Research (BBO), and a Butcher Innovation Award NSF IGERT 1144807 (JRW and TOV), UC Boulder BSI and UROP programs (ONW). RP and HYC are investigators of the Howard Hughes Medical Institute.

# Additional information

## Competing interests

Howard Y Chang: Reviewing editor, *eLife*. Bradley B Olwin: Serves on the scientific board of Satellos Biosciences. The other authors declare that no competing interests exist.

## Funding

| Funder | Grant reference number | Author |
|---|---|---|
| National Institutes of Health | T32GM008497 | Joshua R Wheeler<br>Thomas O Vogler<br>Eric D Nguyen |
| National Institutes of Health | NIH-F30NS093682 | Joshua R Wheeler |
| National Institutes of Health | NIH-F30AR068881 | Thomas O Vogler |
| National Institutes of Health | NIH-GM045443 | Roy Parker |
| National Institutes of Health | NIH-R35GM119575 | Aaron M Johnson |
| National Institutes of Health | NIH-AR049446 | Bradley B Olwin |
| National Institutes of Health | NIH-AR070360 | Bradley B Olwin |
| American Cancer Society | IRG Paul O'Hara II Seed Grant | Aaron M Johnson |
| National Institutes of Health | NIH-RM1-HG007735 | Howard Y Chang |
| National Science Foundation | NSF IGERT 1144807 | Joshua R Wheeler<br>Thomas O Vogler |
| Glenn Foundation for Medical Research | | Bradley B Olwin |
| University of Colorado Boulder | Biological Sciences Initiative | Oscar N Whitney |
| University of Colorado Boulder | University of Colorado Undergraduate Research Program | Oscar N Whitney |
| Howard Hughes Medical Institute | | Howard Y Chang<br>Roy Parker |

The funders had no role in study design, data collection and interpretation, or the decision to submit the work for publication.

## Author contributions

Joshua R Wheeler, Oscar N Whitney, Conceptualization, Data curation, Formal analysis, Investigation, Methodology, Validation, Writing – original draft, Writing – review and editing; Thomas O Vogler, Conceptualization, Investigation, Methodology, Validation, Writing – original draft, Writing – review and editing; Eric D Nguyen, Formal analysis, Investigation, Methodology, Writing – review and editing; Bradley Pawlikowski, Evan Lester, Formal analysis, Investigation, Writing – review and editing; Alicia Cutler, Tiffany Elston, Nicole Dalla Betta, Kathryn E Yost, Hannes Vogel, Investigation, Writing – review and editing; Kevin R Parker, Data curation, Investigation, Writing – review and editing; Thomas A Rando, Howard Y Chang, Aaron M Johnson, Project administration, Supervision, Writing – review and editing; Roy Parker, Bradley B Olwin, Conceptualization, Funding acquisition, Project administration, Supervision, Writing – original draft, Writing – review and editing

## Author ORCIDs

Joshua R Wheeler  http://orcid.org/0000-0002-7315-8269
Oscar N Whitney  http://orcid.org/0000-0002-4858-2615
Thomas O Vogler  http://orcid.org/0000-0002-4537-8248
Alicia Cutler  http://orcid.org/0000-0003-3365-0328
Kathryn E Yost  http://orcid.org/0000-0001-6807-950X
Thomas A Rando  http://orcid.org/0000-0001-5843-8564
Howard Y Chang  http://orcid.org/0000-0002-9459-4393
Aaron M Johnson  http://orcid.org/0000-0003-4553-1078
Roy Parker  http://orcid.org/0000-0002-8412-4152
Bradley B Olwin  http://orcid.org/0000-0001-6977-2509

## Ethics

Mice were bred and housed according to National Institutes of Health (NIH) guidelines for the ethical treatment of animals in a pathogen-free facility at the University of Colorado at Boulder. The University of Colorado Institutional Animal Care and Use Committee (IACUC) approved all animal protocols and procedures and studies complied with all ethical regulations. IACUC protocol number 2516, animal welfare assurance number A3646-01.

## Decision letter and Author response

Decision letter https://doi.org/10.7554/eLife.75844.sa1
Author response https://doi.org/10.7554/eLife.75844.sa2

# Additional files

## Supplementary files

• Transparent reporting form
• Supplementary file 1. Primers used for RIP-qPCR.
• Supplementary file 2. Hnrnpa2b1 eCLIP mapping in Myoblasts and Myotubes.
• Supplementary file 3. Hnrnpa2b1 KO differential gene expression.
• Supplementary file 4. Hnrnpa2b1 KO cluster expression.
• Supplementary file 5.
• Supplementary file 6. RNA-binding proteins expressed in myogenic clusters 0 - 6.

## Data availability

Sequencing data is publicly available (SRA accession: PRJNA717101, GEO accession: GSE152467, GSE106553).

The following datasets were generated:

| Author(s) | Year | Dataset title | Dataset URL | Database and Identifier |
|---|---|---|---|---|
| Wheeler JR, Whitney O, Olwin B, Parker R | 2020 | Hnrnpa21 regulates myogenic fate plasticity | https://www.ncbi.nlm.nih.gov/geo/query/acc.cgi?acc=GSE152467 | NCBI Gene Expression Omnibus, GSE152467 |
| Wheeler JR, Whitney O, Olwin B, Parker R | 2021 | RNA-Binding Proteins Direct Myogenic Cell Fate Decisions | https://www.ncbi.nlm.nih.gov/bioproject/PRJNA717101/ | NCBI BioProject, PRJNA717101 |
| Nguyen ED, Wheeler JR, Vogler TO, Johnson AM | 2022 | RNA-binding proteins direct myogenic cell fate decisions | http://www.ncbi.nlm.nih.gov/geo/query/acc.cgi?acc=GSE106553 | NCBI Gene Expression Omnibus, GSE106553 |

The following previously published datasets were used:

| Author(s) | Year | Dataset title | Dataset URL | Database and Identifier |
|---|---|---|---|---|
| Trapnell C, Williams BA, Pertea G, Mortazavi A, Kwan G, van Baren MJ, Salzberg SL, Wold BJ, Pachter L | 2010 | Transcript assembly and abundance estimation from RNA-Seq reveals thousands of new transcripts and switching among isoforms | https://www.ncbi.nlm.nih.gov/geo/query/acc.cgi?acc=GSE20846 | NCBI Gene Expression Omnibus, GSE20846 |
| Singh RK, Xia Z, Bland CS, Kalsotra A, Ruddy M, Curk T, Ule J, Li W, Cooper TA | 2014 | Rbfox2-coordinated alternative splicing of Mef2d and Rock2 controls myoblast fusion during myogenesis | https://www.ncbi.nlm.nih.gov/geo/query/acc.cgi?acc=GSE58928 | NCBI Gene Expression Omnibus, GSE58928 |
| Thomas JD, Bardhi O, Aslam FN, Sznajder LJ | 2017 | Disrupted prenatal RNA processing and myogenesis in congenital myotonic dystrophy | https://www.ncbi.nlm.nih.gov/geo/query/acc.cgi?acc=GSE97806 | NCBI Gene Expression Omnibus, GSE97806 |

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
