## [Editor Report]

Wheeler and colleagues examine genetic pathways of myogenesis in regenerating muscle. Using extensive single cell and whole-genome analyses, they uncover a new role for the RNA binding protein hnRNP A2B1, showing that it plays a key role in defining muscle-specific splicing patterns during development.

---

## [Decision Letter]

**Decision letter after peer review:**

[Editors’ note: the authors submitted for reconsideration following the decision after peer review. What follows is the decision letter after the first round of review.]

Thank you for submitting your work entitled "RNA-binding proteins direct myogenic cell fate decisions" for consideration by *eLife*. Your article has been reviewed by 3 peer reviewers, and the evaluation has been overseen by a Reviewing Editor and a Senior Editor. The following individual involved in review of your submission has agreed to reveal their identity: Maurice Swanson (Reviewer #3).

Comments to the Authors:

We are sorry to say that, after consultation with the reviewers, we have decided that your work cannot be considered further for publication by *eLife*.

All of the reviewers found the single cell analysis to be a valuable contribution to the field. However, none of the reviewers were convinced by the evidence that hnRNP A2B1 is a key mediator of the myogenic program. These findings were felt to be overstated. It was not clear why this particular factor was chosen for detailed analysis. Given these concerns, it was felt that the manuscript would require very extensive revisions, which would go beyond the revision policy of *eLife*.

*Reviewer #1:*

The authors used single cell sequencing at days 4 and 7 following mouse skeletal muscle injury to identify nine myogenic subclusters. RNA velocity and partition-based graph abstraction were used to identify temporal dynamics and connectivity between clusters. Analysis of RBP mRNA levels shows striking differences between clusters. This is not particularly surprising as one would expect the same if a class of 1500 or, so genes of another category were analyzed. They focus on several RBP known to be involved in disease and specifically Hnrnpa2b1 early during muscle regeneration which they show is increased in Pax7+ cells and in centrally localized nuclei of regenerating myofibers and decline four weeks later. Loss of Hnrnpa2b1 by CRISPR KO in the mouse C2C12 myoblast cell line reduces muscle differentiation (although it is unclear whether whether this result is derived from multiple independent clones or a single clone, and clonal isolates of C2C12 can have a wide variation in differentiation potential) and results in altered splicing of >2000 RNAs including other RNA binding proteins that regulate splicing as well as other RNA processing. To address the possibility that Hnrnpa2b1a is required for myogenic cell population transitions during regeneration, they used a machine learning approach inputting single cell sequence, C2C12 differentiation and Hnrnpa2b1 RNA-seq data. The result suggests a hypothesis that Hnrnpa2b1 splicing function is required for transitions of specific myogenic populations, which unfortunately the authors take as fact. eCLIP analysis of Hnrnpa2b1-associated RNAs in C2C12 proliferating and differentiated cultures was used to identify surprisingly small numbers of genes (247 and 137, respectively) perhaps owing to the high quality of the approach. The data was used to derive a so-called RBP engagement score correlating the expression of the RBP and its target pre-mRNA at the level of single cells. The results were used to support the hypothesis that Hnrnpa2b1 is required for transition from subclusters during myogenic cell transitions.

The results presented are of high quality and when released it will be a very useful data set. The analyses performed lead to reasonable hypothesis but, unfortunately, rarely are they tested and done so directly and rigorously. The authors generated a great deal of data that appears to be of high quality and then tried to make several attempts at connections to RNA binding proteins in human disease, splicing regulation and cross regulation of other RNA binding proteins with Hnrnpa2b1 as a major central regulator of muscle differentiation. Conclusions were often based on inference and correlation or lacked sufficient experimental support.

The results presented are of high quality and when released it will be a very useful data set. One main and in fact disturbing issue with the paper is the superficiality of many of the conclusions and a lack of rigor with regard to making conclusions, a list of only some examples is below. There are many examples of possibilities suggested, implied and what could be the case based on the data, but often the conclusions are based on highly derivative data and the likelihood of coming away with a solid new piece of information is quite weak. The analyses performed lead to reasonable hypothesis but rarely are they tested and done so directly and rigorously to a satisfying endpoint. A few conjectures are reasonable to present; however, some of their major conclusions are very weakly supported and other explanations are just as likely. One example is the conclusions that Hnrnpa2b1 is required for myogenic differentiation and that it is implied to be "a central regulator within a network of splicing factors that directs specific myogenic cell fate". The direct evidence that Hnrnpa2b1 is required for myogenic differentiation comes from CRISPR/Cas9 mediated knock out in C2C12. Clonal Hnrnpa2b1 knock out mouse C2C12 cell lines were generated and Figure 4 shows statistically significant reduced differentiation; however, it is very important to know whether this result is derived from multiple independent clones or a single clone. This information is not available in the manuscript. Clonal isolates of C2C12 will have a wide variation in differentiation potential given the heterogeneity of cells within the parental population. The authors need to demonstrate that the reduced differentiation potential is consistent among several KO cell lines; or show that the phenotype can be rescued by Hnrnpa2b1 expression. Another approach to avoid issues of clonal variability is to use siRNA knock down in a standard C2C12 culture that is known to be efficient throughout the period required for differentiation. With regard to the conclusion that Hnrnpa2b1 is among central regulators, like many statements and conclusions presented, it is quite possible but without a clear demonstration that loss of Hnrnpa2b1 has a consistent physiological effect, it could also be a minor player.

If the authors were to generate a conditional Hnrnpa2b1 KO that was muscle specific and tamoxifen-induced in adult mice (all doable), then did a regeneration assay as in Figure 1, would they find a critical role for Hnrnpa2b1 in regeneration and would they find blockage between subcluster 5 and 6? This would be the definitive experiment to support what is proposed but this is too much to ask.

In the end the technology the single cell Seq, eCLIP etc, are fantastic approaches but are also screens that need to be validated and while I don't understand the computational analysis (and who among the majority of reviewers of such work do), it does require subjective decisions of filtering and criteria selection that is largely opaque so also needs strong experimental validation to produce conclusions that will stand the test of time. In this manuscript I worry that the use of sophisticated technologies and derivative computational analysis supplants definitive experiments, independent validation and well supported conclusions.

*Reviewer #2:*

In this manuscript, the authors used single cell RNA-sequencing, machine learning and other strong techniques to determine the timing of RPB expression during muscle regeneration and identify the networks of RBPs involved in myogenic cell fate transitions.

Authors goal is novel and well framed. Their conclusions are mostly well supported by their extensive set of experiments and deep analysis of their result.

Through the data and analysis presented in the manuscript, the authors achieved their goal. Furthermore, the new information presented here has the potential to be used by several groups with expertise in specific muscle disorders to more deeply understand the mechanism behind those diseases.

Strengths

1. The research in this paper is novel in an area that is under-explored in the splicing field and furthermore it has strong implications in muscle diseases and muscle cell regeneration. I agree with authors' statement: "little is known regarding the mechanisms regulating the timing of RBP function during myogenesis or the mechanisms by which RBPs influence myogenic cell fate decision". Thus, this manuscript has the potential to fill this interesting gap of knowledge in the muscle field.

2. The usage of machine learning is to explore post-transcriptional regulation during cell differentiation is very interesting

3. Using single cell RNA-sequencing in muscle stem cells is a strong novelty and appropriate to achieve the goal of the study.

Weakness

1. It is unclear how the authors discriminate the direct effect of Hnrnpa2b1 on splicing and expression from the fact that cells are not differentiating. This is my main concern since we know that there are extensive changes during normal myogenesis and C2C12 myoblast differentiation into myotubes. I personally think that many, many of the changes that you are detecting are due to the absence of differentiation. I think this aspect / caveat should be discussed and/or rule out.

1. Please, justify some of your statements with references or rationale from your side. One example is in line 169-171: "Although the mechanism of myogenic cell fate change is clearly multifactorial, a likely driver of rapid and dynamic cell fate changes is post-transcriptional control of RNA.". Why? Where are you basing this hypothesis on?

2. Same here (lines 177-178): "While RBP expression increases during muscle regeneration…". All RBPs are upregulated during muscle regeneration?? If so, please, provide the evidence.

3. Figure 4 A-B: please support the microscopy evidence with a more quantitative method for protein expression such as Western blotting. Measuring intensity in microscopy images from immunofluorescence experiments is not an accurate way to quantify levels of protein expression especially when comparing different images. Please, also state which cells are you using in the text and the figure legend. Are those C2C12 cells? I guess so since those cells are used in other panels of Figure 4. But, please, clarify this.

4. How are you discriminating the direct effect of Hnrnpa2b1 on splicing and expression from the fact that myoblasts are not differentiating into myotubes? This is my main concern since we know that there are extensive transcriptional and post-transcriptional changes during in vivo myogenesis and in vitro C2C12 myoblast differentiation into myotubes. I personally think that many of the changes that you are detecting might due to the absence of differentiation.

*Reviewer #3:*

Wheeler et al. use a number of informative tools to investigate the roles of RNA binding proteins (RBPs) in skeletal muscle cell fate decisions initially with scRNAseq of BaCl2 damaged TA muscle 4- and 7-days post injury (dpi) combined with machine learning and RBP engagement scoring to highlight a prominent role for Hnrnpa2b1 as a key myogenic differentiation factor. This conclusion is substantiated by analysis of C2C12 myogenic differentiation in vitro, where the authors unexpectedly discover that this RBP is not essential for myoblast proliferation but is required for myogenic differentiation. Identification of direct downstream splicing targets using RNAseq and eCLIP revealed that Mbnl1, Mbnl2 and Rbfox2, known RBP regulators of myogenesis, were altered in Hnrnpa2b1 KOs suggesting that this RBP is a central splicing and differentiation regulator. To differentiate direct vs indirect effects on splicing, eCLIP is performed and together with RBP engagement scoring and assignment of gene expression signatures to specific clusters indicates a subcluster 5 to 6 block due to Hnrnpa2b1 loss. While previous work has highlighted functions for various RBPs in muscle diseases, this study constitutes a significant addition since it alters our prior views of Hnrnpa2b1 functions during cell proliferation and differentiation, provides insights into stage-specific functions, and details an experimental pathway to similarly classify other ubiquitously expressed RBPs.

The presentation is generally clear but several issues should be addressed.

1. Figures 2A, B and 3. Given the RBP hierarchical clustering and scRNA expression analysis of myogenic clusters, which revealed similar patterns for Hnrnpa1 and Hnrnpa2b1 (except for Subcluster 3), additional justification should be included for the subsequent exclusive focus on Hnrnpa2b1. For example, in Figure 3, do expression characteristics following injury differ between these two RBPs?

2. Figure 4F. Additional information on the differentiation defect in Hnrnpa1b1 KO myoblasts would be helpful – do the KO myoblasts show any morphological changes after differentiation induction?

4. Figure 5B. The exon/intron patterns for Mbnl1, Mbnl2 and Rbfox2 appear to differ significantly from the UCSC browser view (in contrast, Hnrnpa2b1 in Figure S6F/G is correct) so confirm genomic coordinates. In addition, discuss the exons that are up/down in the Hnrnpa2b1 KO and how these alternative splicing patterns might affect the activity or localization of the encoded RBP isoforms.

5. Figure 6A. The majority of Hnrnpa2b1 eCLIP tags are in the 3'UTR (myoblasts, ~47%; myotubes ~65%) so does this suggest that the predominant problem in Hnrnpa2b1 KOs is RNA localization, translation and/or turnover. On p16 (last sentence), note that in myoblasts distal intron binding sites predominate over proximal.

[Editors’ note: further revisions were suggested prior to acceptance, as described below.]

Thank you for submitting your article "RNA-binding proteins direct myogenic cell fate decisions" for consideration by *eLife*. Your article has been reviewed by 3 peer reviewers, and the evaluation has been overseen by Reviewing Editor Douglas Black and Senior Editor James Manley. The reviewers have opted to remain anonymous.

Since this study was previously rejected for publication in *eLife*, this submission is considered a new manuscript.

Wheeler and colleagues investigate the roles of RNA binding proteins in muscle cell fate decisions initially using single cell (sc)RNAseq analysis of injured skeletal muscle at 4- and 7-days post injury. They develop a myogenesis trajectory map and focus on three cell populations that are divided into 9 subclusters defined for connectivity. MuSC RBP expression in subclusters 0-6 revealed high expression of Hnrnpa1 and Hnrnpa2b1. Hnrnpa2b1 levels were seen to increase during regeneration in Pax7+ MuSCs and myofibers. Hnrnpa2b1 knockout clones continue to proliferate but fail to differentiate, and exhibit a large number splicing alterations from wildtype cells, including in the developmental regulators Mbnl1, Mbnl2 and Rbfox2. Direct RNA targets of Hnrnpa2b1 were identified by eCLIP. They use a machine learning tool to estimate myogenic subcluster contributions to bulk RNAseq datasets of proliferating and differentiating myoblasts, with Hnrnpa2b1 loss predicted to lead to subcluster-specific splicing changes.

All of the reviewers found this study to be of interest – providing new insights into RNA metabolism in differentiating muscle and defining stage-specific functions of an important RBP. Nevertheless, a number of issues were raised regarding the data interpretation as well as aspects of the bioinformatics analysis. Addressing these will require additional analyses and other modifications to the study.

Essential revisions

1. There are extensive changes in splicing during normal myogenesis and C2C12 myoblast differentiation into myotubes. The Hnrnpa2b1 C2C12 knockout clones fail to differentiate, so many of the splicing effects may simply reflect failed differentiation and not direct regulation by Hnrnpa2b1. This question was brought up in reviews of the earlier study and is still a concern that is not resolved by the new analyses and bioinformatics. Transcriptional and posttranscriptional changes observed in the Hnrnpa2b1 KO cells should be compared to those known to occur during normal C2C12 cell differentiation (data available in Singh et al., Molecular Cell 2014 PMID: 25087874 and possibly elsewhere). This analysis will allow the authors to discuss how the Hnrnpa2b1 splicing program might differ from a presumably larger program occurring with differentiation.

2. The majority of Hnrnpa2b1 eCLIP tags are in the 3'UTR (myoblasts, ~47%; myotubes ~65%). However, the final model is that Hnrnpa2b1 regulates cell fate decisions via modulation of RNA splicing of both direct targets and alterations in other splicing regulators. It seems likely that other Hnrnpa2b1-mediated events will be equally if not more important. Processes potentially affected by 3' UTR binding include mRNA translation, localization and decay, all of which likely contribute to a differentiation program. The authors should assess whether transcripts bound by Hnrnpa2b1 in their 3' UTR show changes in overall expression rather than splicing in response to Hnrnpa2b1 depletion. Are these 3' UTR binding sites found adjacent to recognition elements for muscle specific microRNAs? Given the many examples of RNA binding proteins of this type affecting differentiation by modulating the processing of pri or pre-miRNAs to mature miRNAs, the authors should examine their existing RNAseq and iCLIP data for evidence of Hnrnpa2b1 affecting miRNA precursors. They should consider profiling the miRNAs in their cells by short RNA sequencing.

3. The genomic data and bioinformatic analyses are not sufficiently described to be adequately assessed.

A. The authors identified over 2000 target RNAs altered in Hnrnpa2b1 KO cells, were these genes or alternative splicing events? It is not clear what degree of confidence can be placed on these findings. How many replicates were profiled? What is the magnitude of the splicing changes? What criteria were used to determine if these changes resulted from direct regulation by Hnrnpa2b1 or were due to indirect effects such as failure to differentiate as described above?

B. The authors point out altered splicing in the splicing regulators Mbnl1/2 and Rbfox2. From the data shown, it is difficult to judge the magnitude of splicing changes, and whether they are sufficient to actually have an impact on their downstream targets.

C. The quality of eCLIP data is unclear. The number of tags before cluster identification is not described in the main text. From the examples shown in Figure S6 F,G, the CLIP tags are widely distributed and somewhat similar to the input. This is an indication of a possible high background in the IP, which may explain the small number of significant peaks when normalized to input. It is unclear whether the binding sites identified by CLIP support splicing changes identified by RNA-seq.

D. Validation of the engagement scoring is not presented. While an RBP and its target need to be coexpressed for a regulatory process to occur, it is not clear how co-expression alone will provide a measure of regulatory activity. For example, if an RBP suppresses gene expression, it might be anticorrelated with its targets.

*Reviewer #1:*

RNA binding proteins (RBPs) play key roles in muscle stem cell (MuSC) quiescence, activation and self-renewal, and disease-associated RBP mutations have been linked to severe muscle loss. In this study, Wheeler and colleagues investigate RBP roles in muscle cell fate decisions initially using single cell (sc)RNAseq analysis of injured skeletal muscle at 4- and 7-days post injury. To define the myogenesis trajectory map they subsequently focus on three cell populations that are further divided into 9 subclusters followed by RNA velocity and partition-based abstraction to highlight inter-cluster connectivity and directionality. MuSC RBP expression in subclusters 0-6 revealed high expression of Hnrnpa1 and Hnrnpa2b1, and since a previous study demonstrated myofibril hypoplasia in mouse Hnrnpa1 knockout mice, they pursued Hnrnpa2b1. Hnrnpa2b1 levels increase during regeneration in Pax7+ MuSCs and myofibers. Multiple Hnrnpa2b1 knockout clones show significant myoblast fusion, but not proliferation, defects and splicing alterations in the developmental regulators Mbnl1, Mbnl2 and Rbfox2. Since the observed splicing changes could result from indirect effects of differentiation failure, they used a machine learning tool to determine myogenic subcluster levels from bulk RNAseq datasets of proliferating and differentiating myoblasts. They find that Hnrnpa2b1 loss leads to subcluster-specific splicing changes, and direct vs indirect RNA targets were identified by eCLIP and RNA engagement scoring. While previous studies have analyzed scRNAseq of regenerating muscle, this study provides novel insights into stage-specific functions of an important RBP and thus provides an experimental pathway to similarly classify additional RBPs. Although prior studies have also analyzed scRNAseq of regenerating muscle, the RBP emphasis and experimental strategy is novel and the observed effects of Hnrnpa2b1 loss on the splicing of other developmental splicing factors is an important contribution. Hnrnpa21 C2C12 knockout clones fail to differentiate, so a remaining concern is that many of the splicing effects may simply reflect failed differentiation although the authors could address this issue by knockout an unrelated gene that is required for C2C12 differentiation.

*Reviewer #2:*

In this manuscript, the authors used single cell RNA-sequencing, machine learning and other novel techniques to determine the timing of RPB expression during muscle regeneration and identify the networks of RBPs involved in myogenic cell fate transitions.

[Editors’ note: further revisions were suggested prior to acceptance, as described below.]

Thank you for resubmitting your work entitled "RNA-binding proteins direct myogenic cell fate decisions" for further consideration by *eLife*. Your revised article has been evaluated by James Manley (Senior Editor) and Douglas Black (Reviewing Editor).

The manuscript has been improved but there are some remaining issues that need to be addressed, as outlined below:

Regarding the correlation of the hnRNP A2B1 binding to splicing regulation, you should consider the standard approach of generating an RBP map that shows clip binding density in relation to a metagene derived from all the up and downregulated exons. There are several tools available for doing this and you should assess whether they can be applied to your data. Such a map would be useful to determine whether splicing activation and repression by hnRNP A2B1 showed similar dependence on binding site position relative the exon as seen with some other regulators, and whether hnRNP A2B1 behaves similarly to its homolog hnRNP A1. In the absence of such a map or in addition, it would also be helpful to show an example or two of a clip peak mapping to a strongly regulated exon. As the data are currently presented it is difficult to assess the Clip in relation to the splicing changes, which your response indicates you want to emphasize.

---

## [Author Response]

[Editors’ note: the authors resubmitted a revised version of the paper for consideration. What follows is the authors’ response to the first round of review.]

Comments to the Authors:We are sorry to say that, after consultation with the reviewers, we have decided that your work cannot be considered further for publication by eLife.All of the reviewers found the single cell analysis to be a valuable contribution to the field. However, none of the reviewers were convinced by the evidence that hnRNP A2B1 is a key mediator of the myogenic program. These findings were felt to be overstated. It was not clear why this particular factor was chosen for detailed analysis. Given these concerns, it was felt that the manuscript would require very extensive revisions, which would go beyond the revision policy of eLife.Reviewer #1:The authors used single cell sequencing at days 4 and 7 following mouse skeletal muscle injury to identify nine myogenic subclusters. RNA velocity and partition-based graph abstraction were used to identify temporal dynamics and connectivity between clusters. Analysis of RBP mRNA levels shows striking differences between clusters. This is not particularly surprising as one would expect the same if a class of 1500 or, so genes of another category were analyzed. They focus on several RBP known to be involved in disease and specifically Hnrnpa2b1 early during muscle regeneration which they show is increased in Pax7+ cells and in centrally localized nuclei of regenerating myofibers and decline four weeks later. Loss of Hnrnpa2b1 by CRISPR KO in the mouse C2C12 myoblast cell line reduces muscle differentiation (although it is unclear whether whether this result is derived from multiple independent clones or a single clone, and clonal isolates of C2C12 can have a wide variation in differentiation potential) and results in altered splicing of >2000 RNAs including other RNA binding proteins that regulate splicing as well as other RNA processing. To address the possibility that Hnrnpa2b1a is required for myogenic cell population transitions during regeneration, they used a machine learning approach inputting single cell sequence, C2C12 differentiation and Hnrnpa2b1 RNA-seq data. The result suggests a hypothesis that Hnrnpa2b1 splicing function is required for transitions of specific myogenic populations, which unfortunately the authors take as fact. eCLIP analysis of Hnrnpa2b1-associated RNAs in C2C12 proliferating and differentiated cultures was used to identify surprisingly small numbers of genes (247 and 137, respectively) perhaps owing to the high quality of the approach. The data was used to derive a so-called RBP engagement score correlating the expression of the RBP and its target pre-mRNA at the level of single cells. The results were used to support the hypothesis that Hnrnpa2b1 is required for transition from subclusters during myogenic cell transitions.The results presented are of high quality and when released it will be a very useful data set. The analyses performed lead to reasonable hypothesis but, unfortunately, rarely are they tested and done so directly and rigorously. The authors generated a great deal of data that appears to be of high quality and then tried to make several attempts at connections to RNA binding proteins in human disease, splicing regulation and cross regulation of other RNA binding proteins with Hnrnpa2b1 as a major central regulator of muscle differentiation. Conclusions were often based on inference and correlation or lacked sufficient experimental support.

We thank the reviewer for their thoughtful comments. As the reviewer raises several points in this comment, for clarity, we have separated out each of these points and will address each below in turn.

The results presented are of high quality and when released it will be a very useful data set. One main and in fact disturbing issue with the paper is the superficiality of many of the conclusions and a lack of rigor with regard to making conclusions, a list of only some examples is below. There are many examples of possibilities suggested, implied and what could be the case based on the data, but often the conclusions are based on highly derivative data and the likelihood of coming away with a solid new piece of information is quite weak. The analyses performed lead to reasonable hypothesis but rarely are they tested and done so directly and rigorously to a satisfying endpoint. A few conjectures are reasonable to present; however, some of their major conclusions are very weakly supported and other explanations are just as likely.

The reviewer's major criticism is many of our conclusions lack experimental validation. We have toned down the language in our manuscript to better align with the presented data.

We would make additional two points in response to the reviewer’s comments.

First, the experimental tools utilized in our manuscript serve to support our conclusions. Our experiments define a role for RNA-binding proteins (RBPs) in directing cell fates. We experimentally confirm the role for one RBP, Hnrnpa2b1, in regulating cell fate decisions. We use a multiomic, genetic, biochemical, and novel computational approaches to support these conclusions. Our work serves as a testament to the power of using a multidisciplinary approach to generate fresh insights into RBP and RNA biology.

Second, we agree that our datasets and methodologies will be of broad interest to the scientific community. Our approach provides a model for the use of multiomic technologies in exploring gene expression regulation during complex biologic processes.

One example is the conclusions that Hnrnpa2b1 is required for myogenic differentiation and that it is implied to be "a central regulator within a network of splicing factors that directs specific myogenic cell fate". The direct evidence that Hnrnpa2b1 is required for myogenic differentiation comes from CRISPR/Cas9 mediated knock out in C2C12. Clonal Hnrnpa2b1 knock out mouse C2C12 cell lines were generated and Figure 4 shows statistically significant reduced differentiation; however, it is very important to know whether this result is derived from multiple independent clones or a single clone. This information is not available in the manuscript. Clonal isolates of C2C12 will have a wide variation in differentiation potential given the heterogeneity of cells within the parental population. The authors need to demonstrate that the reduced differentiation potential is consistent among several KO cell lines; or show that the phenotype can be rescued by Hnrnpa2b1 expression. Another approach to avoid issues of clonal variability is to use siRNA knock down in a standard C2C12 culture that is known to be efficient throughout the period required for differentiation. With regard to the conclusion that Hnrnpa2b1 is among central regulators, like many statements and conclusions presented, it is quite possible but without a clear demonstration that loss of Hnrnpa2b1 has a consistent physiological effect, it could also be a minor player.

The reviewer makes two points here which we address below.

The major criticism is the perceived lack experimental rigor in supporting our conclusions. Here, the reviewer focuses on our knockout experiments as an example of a lack of our experimental rigor. The reviewer contests that three independent clones are necessary to define a triplicate and to make any biologic claim as to the impact of Hnrnpa2b1 loss on splicing changes. The reviewer states, "this information is not available in the manuscript." In two separate sections of our original manuscript, we detailed our generation and use of three biologic clones. First, in the figure legend, we state "All images represent n=3 biological replicates from 3 independent WT and KO Hnrnpa2b1 clones” (Figure S4). Second, in the methods section, we write "Myoblasts KO and WT clones were isolated using cloning ring and selectively detaching clonal populations via trypsinization. Clonal population KO was validated via immunofluorescence and western blotting."

The next issue raised by the reviewer centers on Hnrnpa2b1 impacting RNA splicing. The evidence we provide for this claim is three-fold.

(i) In three separate Hnrnpa2b1 KO clones, loss of Hnrnpa2b1 results in arrest of myogenic differentiation. RNA sequencing shows global splicing alterations in the differentiating Hnrnpa2b1 KO cells compared to wild type. As discussed more thoroughly below, we also show that Hnrnpa2b1 splicing directly impacts specific cell states. Thus, the detected splicing defects are a result of Hnrnpa2b1 functioning as a splicing regulator and are not a result of sequencing populations of cells in different differentiation states.

(ii) Hnrnpa2b1 loss results in the altered splicing for several RBP including Rbfox2, Mbnl1, and Mbnl2. RIP and eCLIP experiments show Hnrnpa2b1 binds to the RNA encoding these RBPs. These results show Hnrnpa2b1 regulates the splicing of myogenic RBPs including Rbfox2, Mbnl1, and Mbnl2. Therefore, Hnrnpa2b1 also directly regulates the splicing of other RBP splicing factors.

(iii) Hnrnpa2b1 knockout results in the loss of function for Rbfox2, Mbnl1, and Mbnl2 due to the altered splicing of key domains in these proteins. The altered splicing results in the exclusion of exons encoding zinc finger domains and RNA recognition motifs involved in RNA binding. We show that the altered spliced RNA populations significantly overlap between Hnrnpa2b1, Rbfox2, Mbnl1, and Mbnl2 KO cell lines. These results are compatible with Hnrnpa2b1 loss resulting in the mis-splicing of Rbfox2, Mbnl1, and Mbnl2 leading to impaired splicing function in these RBPs.

Together, we provide evidence for Hnrnpa2b1 interacting and regulating the splicing of other RBPs including Rbfox2, Mbnl1, and Mbnl2. We further define the timing of Hnrnpa2b1 splicing functionality at a single cell level. These results support our assertion that Hnrnpa2b1 is a central splicing regulator.

If the authors were to generate a conditional Hnrnpa2b1 KO that was muscle specific and tamoxifen-induced in adult mice (all doable), then did a regeneration assay as in Figure 1, would they find a critical role for Hnrnpa2b1 in regeneration and would they find blockage between subcluster 5 and 6? This would be the definitive experiment to support what is proposed but this is too much to ask.

In this comment, the reviewer asks us to make a conditional Hnrnpa2b1 knockout mouse. As the reviewer points out though, “(while) this would be the definitive experiment… this is too much to ask.” We agree that this experiment is beyond the scope of the current manuscript.

We would make two additional points.

First, the complexity and expense of this experiment is unreasonable. The reviewer is requesting a conditional Hnrpa2b1 muscle stem cell specific knockout mouse. Next, the reviewer is asking for single cell RNA sequencing in both conditional knockout and wild type Hnrnpa2b1 mice. Single cell RNA sequencing would be necessary at multiple time points following injury in both wild type and knockout mice. Then, the reviewer is requesting bioinformatic and immunohistochemical validation. This is all to show Hnrnpa2b1 knockout stem cells accumulate in myogenic subcluster 5 and 6. This experimental is unreasonable in its scope and complexity, requiring substantial resources and time. It is unlikely we could complete this request in less than 3 years.

Second, successful completion of this experiment does little to add to the overall conclusions already presented in our manuscript. We show a role for Hnrnpa2b1 in regulating myogenic fate decisions and terminal differentiation. This conclusion is built upon in vitro and in vivo experiments. Further, the reviewer is missing a critical aspect of our manuscript. Our goal is to develop and share a bioinformatic approach which generates insight into the functionality of RBPs using multiomic datasets. Using this approach, we identify and characterize the role for Hnrnpa2b1 in both in vitro and in vivo systems. Our results show the power of our computational approach in generating fresh insight from multiomic data. Additional experiments do little to add to the presented conclusions.

In the end the technology the single cell Seq, eCLIP etc, are fantastic approaches but are also screens that need to be validated and while I don't understand the computational analysis (and who among the majority of reviewers of such work do), it does require subjective decisions of filtering and criteria selection that is largely opaque so also needs strong experimental validation to produce conclusions that will stand the test of time. In this manuscript I worry that the use of sophisticated technologies and derivative computational analysis supplants definitive experiments, independent validation and well supported conclusions.

We appreciate the reviewer's honesty in their review of our manuscript. Here, the reviewer states "I don't understand the computational analysis." We trust the other reviewers have expertise in this area and can/have provided a fair review of our approaches.

We agree with the reviewer that bioinformatic tools serve as a screening method. These methods need experimental validation. Indeed, this is precisely what we show in our manuscript. We show the power of bioinformatics to yield new insight which we confirm using "definitive experiments" and "independent validation."

Reviewer #2:1. Please, justify some of your statements with references or rationale from your side. One example is in line 169-171: "Although the mechanism of myogenic cell fate change is clearly multifactorial, a likely driver of rapid and dynamic cell fate changes is post-transcriptional control of RNA.". Why? Where are you basing this hypothesis on?

We thank the reviewer for this feedback and have supported this statement with literature underscoring the role of splicing defects present in skeletal muscle disease and extensive splicing transitions during skeletal muscle development (Apponi et al., 2011; Hinkle et al., 2019; Weskamp et al., 2021).

2. Same here (lines 177-178): "While RBP expression increases during muscle regeneration…". All RBPs are upregulated during muscle regeneration?? If so, please, provide the evidence.

We thank the reviewer for this feedback and have clarified our statement.

3. Figure 4 A-B: please support the microscopy evidence with a more quantitative method for protein expression such as Western blotting. Measuring intensity in microscopy images from immunofluorescence experiments is not an accurate way to quantify levels of protein expression especially when comparing different images. Please, also state which cells are you using in the text and the figure legend. Are those C2C12 cells? I guess so since those cells are used in other panels of Figure 4. But, please, clarify this.

We thank the reviewer for this feedback and would like to clarify our use of immunofluorescence microscopy for quantitative comparison. All immunofluorescence images taken for quantification were cultured, stained, and imaged in parallel, under identical conditions. Each condition was paired with a control stained with only secondary antibody. Therefore, we feel that our immunofluorescence images are suitable for quantitative comparison. We have clarified this methodology in the methods section. We have also revised all figure legend to identify any images and data referring to C2C12 cells.

4. How are you discriminating the direct effect of Hnrnpa2b1 on splicing and expression from the fact that myoblasts are not differentiating into myotubes? This is my main concern since we know that there are extensive transcriptional and post-transcriptional changes during in vivo myogenesis and in vitro C2C12 myoblast differentiation into myotubes. I personally think that many of the changes that you are detecting might due to the absence of differentiation.

The reviewer is inquiring about the splicing changes in Hnrnpa2b1 knockout cells. The observed splicing changes may be due to Hnrnpa2b1 regulating splicing. Or the splicing changes may be due to sequencing two different cell populations. We agree with the reviewer that this is an important stipulation of our data. We have changed the language in our manuscript to reflect this possibility. In addition, we have performed analyses detailed below which show Hnrpa2b1 is a direct RNA splicing regulator.

To control for population-wide differences, we revisited our CIBERSORT analysis. This analysis defines the single cell cluster abundances in both the knockout and wild type cell populations. Cluster 2, cluster 3, cluster 4, cluster 5, and cluster 7 comprise both the knockout and wild type cell populations (See Figure 5E). We next performed differential gene expression analysis for each of these clusters. Differentially expressed genes are enriched in the cells in these particular clusters. This provides a molecular footprint for the cells at these particular stages. We examined the splicing changes for these differentially expressed genes. We find that Hnrnpa2b1 loss leads to splicing alterations in each of these clusters with cluster 5 showing the most splicing changes. This analysis allows us to control for population-wide cell state differences. The results support our contention that Hnrnpa2b1 is a key splicing regulator in particular cell states.

We would further note this analysis shows the power of our bioinformatic approach in defining cell state phenotypes. We expect this approach to be of broad interest to the developmental biology field provided the historic challenge with studying differentiation phenotypes.

**Author response image 1. sa2fig1:** Impact of Hnrnpa2b1 KO on mRNA splicing of myogenic cluster-specific differentially expressed mRNA transcripts. (**A**) and (**B**) Differential gene expression analysis was performed on individual single cell myogenic clusters to identify significantly differentially expressed transcripts. The presence of splicing alterations identified in Hnrnpa2b1 KO cells were assessed in the significantly differentially expressed transcripts. Presented is the percentage of splicing alteration in significantly differentially expressed transcripts identified per myogenic cluster.

Reviewer #3:The presentation is generally clear but several issues should be addressed.1. Figures 2A, B and 3. Given the RBP hierarchical clustering and scRNA expression analysis of myogenic clusters, which revealed similar patterns for Hnrnpa1 and Hnrnpa2b1 (except for Subcluster 3), additional justification should be included for the subsequent exclusive focus on Hnrnpa2b1. For example, in Figure 3, do expression characteristics following injury differ between these two RBPs?

We have provided additional justification for our focus on Hnrnpa2b1.

2. Figure 4F. Additional information on the differentiation defect in Hnrnpa1b1 KO myoblasts would be helpful – do the KO myoblasts show any morphological changes after differentiation induction?

We conducted additional analyses examining for morphological changes. We analyzed the area and circularity of Hnrnpa2b1 wild type and KO myoblasts during differentiation (Supplementary Figure 4E and F). Here, we find minor differences in nuclear area and circularity between differentiating Hnrnpa2b1 WT and KO myoblasts. We have commented on these changes in the Results section.

4. Figure 5B. The exon/intron patterns for Mbnl1, Mbnl2 and Rbfox2 appear to differ significantly from the UCSC browser view (in contrast, Hnrnpa2b1 in Figure S6F/G is correct) so confirm genomic coordinates. In addition, discuss the exons that are up/down in the Hnrnpa2b1 KO and how these alternative splicing patterns might affect the activity or localization of the encoded RBP isoforms.

We thank the reviewer for their astute comments. Here, the reviewer makes two points which we separate out below.

In the first point, the reviewer comments that gene plots for Mbnl1, Mbnl2, and Rbfox2 are different than the UCSC web browser. We appreciate the reviewer bringing this to our attention. The splicing sashimi plots presented in Figure 5B are based on the GENCODE data from GRCm38/mm10. We have validated the splicing coordinates for each of these genes in the UCSC web browser and find that they accurately map to GRCm38/mm10. We have updated the Figure legend to reflect this.

In the second comment, the reviewer asks for a discussion of the exons that are “up/down in the Hnrpa2b1 KO and how these alternative splicing may affect” these transcripts. We have included a discussion of these findings specifically for the impact of splicing alterations on Mbnl1, Mbnl2, and Rbfox2.

5. Figure 6A. The majority of Hnrnpa2b1 eCLIP tags are in the 3'UTR (myoblasts, ~47%; myotubes ~65%) so does this suggest that the predominant problem in Hnrnpa2b1 KOs is RNA localization, translation and/or turnover. On p16 (last sentence), note that in myoblasts distal intron binding sites predominate over proximal.

The issue raised by the reviewer is whether Hnrnpa2b1 knockout impacts 3’UTR site selection. We performed additional analyses examining 3'UTR site selection in Hnrnpa2b1 knockout cells. In this analysis, we quantified 3'UTR site section using QAPA (Ha et al., 2018). QAPA provides a calculation for relative usage of alternative 3′ UTR isoforms based on transcript abundance. Our results show Hnrnpa2b1 loss has minimal impact on 3’UTR site selection (see Author response image 2). Thus, while Hnrnpa2b1 binds to the 3’-end of target RNAs, our results show Hnrnpa2b1 has little impact on 3’UTR location. As the reviewer rightfully points out, Hnrnpa2b1 3’-interactions may be related to Hnrnpa2b1 influencing RNA localization or decay. We have edited the text accordingly.

**Author response image 2. sa2fig2:** Hnrnpa2b1 KO has minimal effect on alternative polyadenylation site selection. (**A**) Analysis of alternative polyadenylation (APA) from Hnrnpa2b1 wild type and KO RNA-seq data using QAPA (Quantification of Alternative Polyadenylation, HA et al., 2018) reveals loss of Hnrnpa2b1 results in no charge in the 3’UTR site selection for the majority of mRNA transcripts.

[Editors’ note: what follows is the authors’ response to the second round of review.]

The reviewers have discussed their reviews with one another, and the Reviewing Editor has drafted this to help you prepare a revised submission.Essential revisions1. There are extensive changes in splicing during normal myogenesis and C2C12 myoblast differentiation into myotubes. The Hnrnpa2b1 C2C12 knockout clones fail to differentiate, so many of the splicing effects may simply reflect failed differentiation and not direct regulation by Hnrnpa2b1. This question was brought up in reviews of the earlier study and is still a concern that is not resolved by the new analyses and bioinformatics. Transcriptional and posttranscriptional changes observed in the Hnrnpa2b1 KO cells should be compared to those known to occur during normal C2C12 cell differentiation (data available in Singh et al., Molecular Cell 2014 PMID: 25087874 and possibly elsewhere). This analysis will allow the authors to discuss how the Hnrnpa2b1 splicing program might differ from a presumably larger program occurring with differentiation.

The issue is whether the gene expression and splicing changes are due to impaired differentiation or due to Hnrnpa2b1 loss. The reviewer asks for gene expression and splicing analysis of proliferating and differentiating C2C12s. This analysis will identify expression and splicing changes associated with differentiation. These expression changes are then compared to the gene and splicing changes seen in Hnrna2bp1 knockout. Hnrnpa2b1-specific splicing changes denote an Hnrnpa2b1 regulatory program distinct from normal differentiation-induced changes.

We completed the requested analysis (See Figure 1). Data presented herein can be incorporated as supplemental data upon request.

RNA sequencing datasets with comparable sequencing depth, quality, and differentiation growth conditions were selected and compared to Hnrnpa2b1 wild type and knockout cell lines (Liu et al., 2020). While gene expression changes may arise due to different growth or sequencing conditions, the differentiating datasets correlate between detected RNA transcripts and gene expression changes (Figure 1A-B). Thus, the datasets can be meaningfully compared.

We next performed differential gene expression and differential splicing analysis (Figure 1C-D). We compared each dataset to a 0hr proliferating myoblast population. These results identify transcripts which are differentially expressed or spliced because of differentiation. Most differentially expressed and spliced transcripts overlap. These shared transcripts likely represent the larger differentiation-induced gene expression program. A subset of unique splicing and expression changes are present in each condition. These unique changes may arise due to differences in growth conditions and heterogeneity of timing for differentiation. In Hnrnpa2b1 knockout (KO) cells a much larger number of unique differentially expressed and alternatively spliced transcripts is present. Thus, the Hnrnpa2b1-KO-specifc transcripts denote a gene expression and splicing program distinct from normal differentiation.

Consistent with our prior observations, a subset of Hnrnpa2b1-KO-specifc alternatively spliced transcripts are differentially expressed (Figure 1E). Hnrnpa2b1-KO-specifc alternatively spliced transcripts are also overrepresented in select myogenic clusters (Figure 1F). These results are consistent with our original results positing an Hnrnpa2b1-cell state specific function during muscle differentiation.

Finally, we note while CIBERSORTx deconvolution show variations in differentiation dynamics between the wild type and Hnrnpa2b1-KO, the differences are small with respect to the total population (Manuscript Figure 5D). Thus, most gene expression changes likely reflect an Hnrnpa2b1-centric regulatory program.

2. The majority of Hnrnpa2b1 eCLIP tags are in the 3'UTR (myoblasts, ~47%; myotubes ~65%). However, the final model is that Hnrnpa2b1 regulates cell fate decisions via modulation of RNA splicing of both direct targets and alterations in other splicing regulators. It seems likely that other Hnrnpa2b1-mediated events will be equally if not more important. Processes potentially affected by 3' UTR binding include mRNA translation, localization and decay, all of which likely contribute to a differentiation program.

We agree with the reviewer. While we chose to focus on the impact of Hnrnpa2b1 on RNA splicing, Hnrnpa2b1 likely has diverse roles in RNA regulation. This is a point acknowledged in the manuscript.

The main issue raised regards the function of Hnrnpa2b1 binding to target RNA 3’UTR’s. These functions include RNA stabilization, splicing, poly-adenylation site selection, and miRNA interaction. To decouple these from each other, we performed analyses examining the effects on Hnrnpa2b1 3’UTR-bound RNAs (See Author response images 3 and 4).

Altogether, our results identify RNA splicing regulation as a major function of Hnrnpa2b1 during myogenesis. As we cannot exclude the influence of Hnrnpa2b1 3’UTR interaction on RNA localization or translation, we have discussed these possibilities in the manuscript.

The authors should assess whether transcripts bound by Hnrnpa2b1 in their 3' UTR show changes in overall expression rather than splicing in response to Hnrnpa2b1 depletion.

We completed the requested analysis (Author response image 3). The majority (81%) of Hnrnpa2b1-bound 3’UTRs show no change in expression in Hnrnpa2b1-knockout cells.

**Author response image 3. sa2fig3:** Hnrnpa2b1-KO shows limited impact on gene expression or poly-A site selection for 3’UTR-Hnrnpa2b1 bound transcripts. (**A**) MA plot of Hnrnpa2b1-3’UTR bound RNA expression in Hnrnpa2b1-KO cells (**B**) Distribution of significantly differentially expressed transcripts in Hnrnpa2b1-KO cells. (**C**) QAPA analysis of Hnrnpa2b1-KO showing impact on 3’UTR and poly-A site selection.

We also examined the role of Hnrnpa2b1-KO on alternative 3’UTR use and poly-adenylation selection. These results show no significant difference in Hnrnpa2b1-KO on alternative 3’UTR use and poly-adenylation selection. For completeness, we also include this analysis (Author response image 3).

Given the many examples of RNA binding proteins of this type affecting differentiation by modulating the processing of pri or pre-miRNAs to mature miRNAs, the authors should examine their existing RNAseq and iCLIP data for evidence of Hnrnpa2b1 affecting miRNA precursors. They should consider profiling the miRNAs in their cells by short RNA sequencing.

We completed the requested analysis (Author response image 4). The majority (98%) of miRNA precursors show no significant change in expression Hnrnpa2b1-knockout cells.

**Author response image 4. sa2fig4:** Impact of Hnrnpa2b1-KO on pre-miRNA expression and 3’UTR interaction. (**A**) MA plot for pre-miRNA expressed in Hnrnap2b1-KO cells. (**B**) Percentage of significantly differentially expressed pre-miRNA in Hnrnap2b1-KO cells. (**C**) Venn diagram showing overlap between Hnrnpa2b1-3’UTR target RNAs and target genes for pre-miRNAs detected in Hnrnpa2b1-WT/KO RNA-sequencing. (**D**) Venn diagram showing overlap between Hnrnpa2b1-3’UTR target RNAs and differentially expressed miRNAs target genes (Zhou et al., 2020).

The reviewer is also requesting for small RNA sequencing on our Hnrnpa2b1-WT and KO cell populations. The sequencing depth of 100-200 million reads per library in our Hnrnpa2b1-WT and KO cell populations is necessary for detecting intronic reads to permit accurate splicing analysis. At this sequencing depth, even lowly expressed pre-miRNA precursors are comprehensively profiled by our sequencing.

Are these 3' UTR binding sites found adjacent to recognition elements for muscle specific microRNAs?

The reviewer is asking about the proximity of microRNAs to Hnrnpa2b1 3’UTR RNA binding sites. To address this point, we performed two additional analyses.

First, we used miwalk to map target genes for the pre-miRNA detected in our sequencing datasets (Sticht et al., 2018). Miwalk-identified miRNA target genes were then compared to all Hnrnpa2b1-3’UTR transcripts. No overlap is seen between pre-miRNA targets and Hnrnpa2b1-3’UTR targets (Author response image 4).

Next, we performed differential gene expression analysis for small RNA sequencing performed on 0-hr and 48-hr in differentiating C2C12 cells (Zhou et al., 2020). Target gene interactions for significantly differentially expressed 48-hr miRNAs were predicted using miwalk (Sticht et al., 2018). The miRNA bound genes were then compared to Hnrnpa2b1-3’UTR target transcripts. Again, no overlap is seen in differentially expressed miRNA and Hnrnpa2b1-3’UTR transcripts (Figure 3D).

The limited change in pre-miRNA expression in Hnrnap2b1-KO cells and absence of co-target transcript binding argues Hnrnpa2b1 has a limited impact on miRNA biosynthesis and function.

3. The genomic data and bioinformatic analyses are not sufficiently described to be adequately assessed.

The reviewer makes several points and requests several additional analyses. We separate out each point and discuss below. Analysis results are presented in Figure 6—figure supplement 2 and Author response image 5.

A. The authors identified over 2000 target RNAs altered in Hnrnpa2b1 KO cells, were these genes or alternative splicing events? It is not clear what degree of confidence can be placed on these findings. How many replicates were profiled? What is the magnitude of the splicing changes? What criteria were used to determine if these changes resulted from direct regulation by Hnrnpa2b1 or were due to indirect effects such as failure to differentiate as described above?

**Author response image 5. sa2fig5:** Magnitude and spatial location for significant Hnrnap2b1-KO splicing alterations. Δ PSI and p-value for significant splicing changes detected in MBNL1/2 and Rbfox2.

The first issue raised regards the number of splicing changes detected. To clarify, a total of 12,292 splicing changes are present in Hnrnpa2b1-KO cells. Of these, 2,571 splicing changes are significant (FDR<0.05) across a total of 2,167 genes. We have edited the text to make this point clearer.

The next point is the degree of confidence and criteria used for detecting splicing changes. The algorithms used to determine splicing alterations are well-established. These algorithms are amongst the best for detecting splicing changes from short read sequencing (Li and Knowles et al., 2018). Briefly, Leafcutter splicing analysis first identifies alternative excised introns from split-reads. These introns must have at least 6 nucleotides mapping into each exon and greater than 30 detected reads across all samples. A Dirichlet-Multinomial generalized linear model is then applied. This model allows for differential intron excision between two groups. A Benjamini-Hochberg FDR is then calculated to identify significantly differentially excised introns. This analysis permits high-confidence identification of splicing changes from short read sequencing.

The reviewer inquires about the number of replicates profiled. We generated three Hnrnpa2b1 knockout and wildtype C2C12 clones. We profiled each clonal population with deep RNA sequencing (>100 million reads per library). Thus, all data represents three independent biologic replicates per condition. We have clarified this point in the methods section and thank the reviewer for bringing this point to our attention.

Next, the reviewer asks us to examine the size of detected splicing changes. We have performed the requested analysis by plotting δ-PSI versus p-value (FDR) (Figure 6—figure supplement 3A).

Finally, the reviewer queries whether the splicing changes are a result of Hnrpa2b1 knockout, or are due to changes in differentiation state. We refer the reviewer to comment #1 and Figure 5—figure supplement 2 for a thorough discussion and associated results addressing this point.

B. The authors point out altered splicing in the splicing regulators Mbnl1/2 and Rbfox2. From the data shown, it is difficult to judge the magnitude of splicing changes, and whether they are sufficient to actually have an impact on their downstream targets.

The issue raised by the reviewer is whether the observed splicing changes in Mbnl1/2 and Rbfox2 are sufficient to cause altered splicing.

Two observations suggest that splicing changes in Mbnl1/2 and Rbfox2 result in altered function. First, significant splicing changes in Mbnl1/2 and Rbfox2 center on key RNA-binding protein domains (Manuscript Figure 5B). Second, Hnrnpa2b1-KO, Mbnl1/2-KO and Rbfox2-KO show significant overlap in spliced transcripts (Manuscript Figure 5C and S5A). These results strongly implicate impaired splicing functionality in Mbnl1/2 and Rbfox2. We have also plotted δ-PSI versus p-value (FDR) for significant splicing changes in Mbnl1/2 and Rbfox2 (Author response image 5).

C. The quality of eCLIP data is unclear. The number of tags before cluster identification is not described in the main text. From the examples shown in Figure S6 F,G, the CLIP tags are widely distributed and somewhat similar to the input. This is an indication of a possible high background in the IP, which may explain the small number of significant peaks when normalized to input. It is unclear whether the binding sites identified by CLIP support splicing changes identified by RNA-seq.

The reviewer is inquiring about the quality of our eCLIP data.

The advantage of using eCLIP is the inclusion of a size matched input (SMinput) control (van Nostrand et al., 2016). The size mapped input reflects all RNA-protein complexes which migrate in the desired size range. As such, a size matched input, IG-IP control, and Hnrnpa2b1-IP are all sequenced and mapped. Hnrnpa2b1 peaks are then normalized against size matched input by calculating fold enrichment of reads in IP versus input. Significant peaks are identified if the number of reads in the IP sample was greater than in the input sample, with a Bonferroni corrected Fisher exact p-value less than 10^–8^. The small number of significant peaks identified is thus a reflection of the high stringency of the eCLIP approach. The use of SMinput, strict p-value cut-off, and the exclusion of peaks identified in IgG-IP control provide high confidence for Hnrnpa2b1-bound RNAs.

The reviewer also contests the eCLIP peak distribution across the Hnrnpa2b1 transcript is a sign of high background. Indeed, sequencing peaks distribute across the transcripts in Manuscript Figure S6F-G. However, dark colored boxes are also shown in this figure. The dark boxes denote significantly enriched binding sites above SMInput (p-value less than 10^–8^). These binding sites center on Hnrnpa2b1’s own 3’UTR, which are previously identified interaction targets for Hnrnpa2b1. We apologize for not making this point clearer and have updated the text to reflect this.

It is unclear whether the binding sites identified by CLIP support splicing changes identified by RNA-seq.

Here, the reviewer is asking how Hnrnpa2b1 eCLIP binding sites relate to splicing changes. If Hnrnpa2b1 binds to target RNAs, then Hnrnpa2b1 may regulate the splicing of target transcripts. Thus, splicing changes and binding sites should be proximally related. To test this, we performed distance and proximity mapping analyses (Author response image 4).

First, we examined the location of splicing changes and eCLIP binding sites. Here, we mapped the locations of splicing clusters and eCLIP peaks using bedtools. We then determined the number of splicing or eCLIP peaks overlapping each gene. We subdivided each splicing cluster and eCLIP peak into percentiles across each gene to compare events across all target genes. The results show proximity of Hnrnpa2b1 eCLIP binding sites to splicing alterations (Figure 6—figure supplement 2B).

Second, we performed relative distance mapping for eCLIP and splicing clusters. These results show eCLIP peaks are more likely to be proximal to an alternative splicing change than by random chance (Figure 6—figure supplement 2C).

These results show Hnrnpa2b1 target RNA binding and splicing changes are proximally related. Thus, Hnrnpa2b1 interaction contributes to the splicing of target transcripts.

D. Validation of the engagement scoring is not presented. While an RBP and its target need to be coexpressed for a regulatory process to occur, it is not clear how co-expression alone will provide a measure of regulatory activity. For example, if an RBP suppresses gene expression, it might be anticorrelated with its targets.

The reviewer makes two points in this comment.

First, the reviewer asks for validation of engagement scoring.

We believe the data presented provides sufficient validation for engagement scoring. Hnrnpa2b1 engagement scoring shows a higher score in differentiating myogenic cluster 5 (Manuscript Figure 6D). A high engagement score suggests higher Hnrnpa2b1 functionality in those clusters. Thus, Hnrnpa2b1-KO would impact clusters 5. Indeed, machine learning shows Hnrnpa2b1-KO leads to increased number of cells arrested in myogenic cluster 5 (Manuscript Figure 5D). And, by immunofluorescent staining, we detect an increase in cells enriched in myogenic cluster 5 (Manuscript Figure 6E-G). Finally, we show Hnrnpa2b1-KO spliced transcripts impact myogenic cluster 5 transcripts (Manuscript Figure 5E and Figure 5—figure supplement 2F). These results provide computational and experimental validation for engagement scoring.

The second point relates to the limits of engagement scoring. The reviewer asserts a limitation is the reliance on co-expressed RBPs and transcripts. We concur. Indeed, engagement scoring relies on the principle that target RNAs and RBP are detected. In our analysis we are examining the expression of an RBP and its RNA targets across all cells within a cluster. We recognize not every cell will express a target RNA either due to limits of detection or due to transcriptional absence. We note that engagement scoring aggregates RNA target expression across all cells within a cluster. Thus, the absence of select transcripts has a limited impact on the ability to assign RBP cluster functionality. In this regard, our method is analogous to assigning transcription factor functionality in single cells (Schep et al., 2017). These methods permit the assessment of functionality in the setting of sparse data points.

[Editors’ note: further revisions were suggested prior to acceptance, as described below.]

The manuscript has been improved but there are some remaining issues that need to be addressed, as outlined below:Regarding the correlation of the hnRNP A2B1 binding to splicing regulation, you should consider the standard approach of generating an RBP map that shows clip binding density in relation to a metagene derived from all the up and downregulated exons. There are several tools available for doing this and you should assess whether they can be applied to your data. Such a map would be useful to determine whether splicing activation and repression by hnRNP A2B1 showed similar dependence on binding site position relative the exon as seen with some other regulators, and whether hnRNP A2B1 behaves similarly to its homolog hnRNP A1. In the absence of such a map or in addition, it would also be helpful to show an example or two of a clip peak mapping to a strongly regulated exon. As the data are currently presented it is difficult to assess the Clip in relation to the splicing changes, which your response indicates you want to emphasize.

We have provided the latter as we have extensive information regarding the splice sites already included in the manuscript and we have revised the information to make this clearer to the readers. See Figure 5 Supplement 2, Figure 6 Supplement 2 and Figure 6 Supplement 3.

We find that *Hnrnpa2b1* and *Myl1* show the most significant splicing alterations and harbor eCLIP binding sites. We have added an additional supplement to demonstrate the requested mapping. The data are now Figure 6 Supplement 3.

We have revised the manuscript to address the requested changes and hope that the reviewers are satisfied with our responses. The revised manuscript, revised figures and supplements are included.